# Data Market Design through Deep Learning

**Sai Srivatsa Ravindranath**[*]      **Yanchen Jiang**[*]      **David C. Parkes**
Harvard John A. Paulson School of Engineering and Applied Sciences
{saisr, yanchen_jiang, parkes} @g.harvard.edu

## Abstract

The *data market design* problem is a problem in economic theory to find a set of signaling schemes (statistical experiments) to maximize expected revenue to the information seller, where each experiment reveals some of the information known to a seller and has a corresponding price [7]. Each buyer has their own decision to make in a world environment, and their subjective expected value for the information associated with a particular experiment comes from the improvement in this decision and depends on their prior and value for different outcomes. In a setting with multiple buyers, a buyer's expected value for an experiment may also depend on the information sold to others [12]. We introduce the application of deep learning for the design of revenue-optimal data markets, looking to expand the frontiers of what can be understood and achieved. Relative to earlier work on deep learning for auction design [32], we must learn signaling schemes rather than allocation rules and handle *obedience constraints*—these arising from modeling the downstream actions of buyers—in addition to incentive constraints on bids. Our experiments demonstrate that this new deep learning framework can almost precisely replicate all known solutions from theory, expand to more complex settings, and be used to establish the optimality of new designs for data markets and make conjectures in regard to the structure of optimal designs.

## 1 Introduction

Many characterize the current era as the Information Age. Companies such as Google and Meta (search and social media), Experian and TransUnion (credit agencies), and Amazon and American Express (commerce) hold vast quantities of data about individuals. In turn, this has led to data markets, where information about an individual can be purchased in real-time to guide decision-making (e.g., LiveRamp, Segment, Bloomreach). In this paper, we advance the design of market rules that govern the structuring and sale of this kind of information. Using machine learning, we leverage existing theoretical frameworks [7, 12] to replicate all known solutions from theory, expand to more complex settings, establish the optimality of new designs, and make conjectures in regard to the structure of optimal designs. Although this is not our focus, there are also important ethical questions in regard to data markets, in regard to privacy, ownership, informed consent, and the use of data [2, 11, 8, 5, 19, 33, 1].

In settings with a single buyer, Bergemann et al. [7] introduce a framework in which there is a *data buyer* who faces a decision problem under uncertainty, and has a payoff depending on their choice of action and the underlying state of the world. There is a *data seller* who knows the world state, and can disclose information through a menu of statistical experiments, each experiment offering a stochastic signal that reveals some (or all) of the seller's information and each of which is associated with a price. The buyer's willingness to pay for information is determined by their *type*, which defines their prior belief and their value for different outcomes in the world (these arising from the action that the buyer will choose to take).

---

[*]Equal Contribution.

37th Conference on Neural Information Processing Systems (NeurIPS 2023).

The *optimal data market design problem* is to find a set of experiments and associated prices with which to maximize expected revenue, given a distribution over the buyer's type. Bergemann et al. [7] characterize the optimal design when there is a binary world state and a buyer with a binary action. In some settings there may be multiple buyers and buyers may compete downstream, and the more informed one buyer is, the lower the payoff may become for others. Bonatti et al. [12] take up this problem, and characterize the optimal design for multiple buyers in the binary state/binary action setting, and further limited to buyers who each have a common prior on the world state. It remains an open problem to obtain theoretical results for richer, multi-buyer settings, and this motivates the need for gaining new understanding and making progress through computational approaches.

**Contributions.** Inspired by the recent advances in *differential economics* ([32, 34, 35, 38, 46, 47, 44, 45, 23, 25, 28, 36, 24, 29] etc.), we initiate the use of deep learning for the automated design of data markets. This market sells information rather than allocates resources, and the value of information depends on the way in which a buyer will use the information—and this downstream action by a buyer needs to be modeled. Following the *revelation principle for dynamic games* ([43], Section 6.3), it is without loss of generality to model an experiment as generating a recommended action and insisting on designs in which the buyer will prefer to follow this action. This brings about new challenges, most notably to extend this framework to handle *obedience*. While the other aspect of incentive compatibility that we need to handle is more typical, i.e., that of achieving incentive alignment so that a buyer will choose to truthfully report their prior beliefs and values for different outcomes, we also need to ensure there are no useful *double deviations*, where a buyer simultaneously misreports their type and acts contrary to the seller's recommended action.

In settings with a single buyer, we learn an explicit, parameterized representation of a menu of priced experiments to offer the buyer and with which to model the action choices available to the buyer. In this way, we extend the *RochetNet architecture* [32] that has been used successfully for optimal auction design with a single buyer. This enables us to obtain exact incentive compatibility for the single buyer setting: a buyer has no useful deviation from a recommended action, no useful deviation in reporting their type, and no useful double deviation. In settings with multiple buyers, we seek to learn revenue-maximizing designs while also minimizing deviations in disobeying action recommendations and misreporting types and including double deviations. This extends the *RegretNet framework* [32] that has been used successfully for optimal auction design with multiple buyers and gives approximate incentive alignment.

Our first experimental result is to show through extensive experiments that these new neural network architectures and learning frameworks are able to almost exactly recover all known optimal solutions from Bergemann et al. [7] and Bonatti et al. [12]. Following the economic theory literature on data market design, we consider the notion of Bayesian incentive compatibility (BIC). To handle this, we use samples of a buyer's type to compute an *interim* experiment and *interim* payment, averaging over samples drawn from the type distribution (this builds on the BIC methods for deep learning for auctions that were used in the context of differentiable economics for budget-constrained auctions [34]). We give a training method that enables the efficient reuse of computed *interim* allocations and *interim* payments from other samples to swiftly calculate the *interim* utility of misreports, dramatically speeding up training.

Whereas analytical results are only available for the BIC setting, which is, in effect, lower-dimensional, and easier to analyze, we are able to study through computational techniques the design of data markets in the *ex post IC setting*, which is a setting without existing theory. A second result, in the setting of multiple buyers, is to use our framework to conjecture the structure of an optimal design and prove its optimality (for this, we make use of *virtual values* analogous to Myerson's framework [42]). We see this as an important contribution, as *ex post IC* is a stronger notion of IC than BIC.

A third result is to demonstrate how our framework extends its utility beyond empirical results and serves as a toolbox to guide economic theory. To illustrate this, we study how revenue varies with competition in the multi-buyer setting where the prior information is uncertain. Despite the absence of existing theoretical results for this particular setting, our framework enables us to derive trends in revenue effortlessly. We also conjecture the structure of solutions for problems in the single buyer setting with an enlarged type, where both the buyer payoffs and priors are uncertain. For this case, we again derive empirical results using our proposed framework and use it to conjecture the properties of the underlying theoretical solution.

**Related work.** Conitzer and Sandholm [20, 21] introduced the use of *automated mechanism design (AMD)* for economic design and framed the problem as an integer linear program (or just a linear program for Bayesian design). Responding to challenges with the enumerative representations used in these early approaches, Dütting et al. [32] introduced the use of deep neural networks for auction design, attaining more representational flexibility. Since then, there has been a line of work on this so-called approach of *differentiable economics*, including to problems with budget-constrained bidders [34], for minimizing agents' payments [47], applying to multi-facility location problems [35], balancing fairness and revenue ([38]), and applying to two-sided matching [46]. To the best of our knowledge, none of this work has considered the setting of the design of optimal data markets, which introduce new challenges in regard to handling agent actions (obedience) as well as incorporating negative externalities.

In regard to the data market problem, Bergemann et al. [7] build upon the decision-theoretic model pioneered by Blackwell [10] and study a setting with a single buyer. Cai and Velegkas [13] also give computational results in this model, making use of linear programs to compute the optimal menu for discrete type distributions. They also investigate a generalization of this model that allows multiple agents to compete for useful information. In this setting, at most, one agent receives the information. While this approach can be used for continuous type distribution by applying the LP to a discretized valuation space, solving it for even a coarser discretization can be prohibitively expensive. Further advancing economic theory, Bergemann et al. [9] consider more general type distributions and investigate both the cardinality of the optimal menu and the revenue achievable when selling complete information. Bonatti et al. [12] also study the multi-buyer setting modeling competition through negative externality.

Babaioff et al. [6] give a related framework, with the key distinction that the seller is not required to commit to a mechanism before the realization of the world state. As a result, the experiments and prices of the data market can be tailored to the realized state of the world. Chen et al. [18] extend their setting, considering budget-constrained buyers, formulating a linear program to find the solution of the problem for a discrete type space.

There exist several other models that study revenue-optimal mechanisms for selling data. For example, Liu et al. [40] characterize the revenue-optimal mechanism for various numbers of states and actions, and considering general payoff functions. However, in their setup, the state only impacts the payoff of the active action taken by a buyer, which provides considerable simplification and may not be realistic. Li [39] investigates a setting where the buyer can conduct their own costly experiment at a cost after receiving the signal. Different from the model considered in this paper, after receiving the signal from the data broker, the agent can subsequently acquire additional information with costs. The model also assumed that the valuation function of the agent is separable and that the private type of the agent represents her value of acquiring more information, which is different from the single buyer model studied in this paper, where the prior belief is also drawn from a distribution and constitutes a part of the buyer's type. Agarwal et al. [3] explore data marketplaces where each of multiple sellers sells data to buyers who aim to utilize the data for machine learning tasks, and Chen et al. [17] consider scenarios where neither the seller nor the buyer knows the true quality of the data. Mehta et al. [41] and Agarwal et al. [4] also incorporate buyer externalities into the study of data marketplaces.

Another line of research studies problems of *information design*, for example, the problem of *Bayesian Persuasion* [37]. There, the model is different in that the sender of information has preferences on the action taken by the receiver, setting up a game-theoretic problem of strategic misrepresentation by the sender. Dughmi and Xu [31] studied this from an algorithmic perspective, and Castiglioni et al. [16] also brought in considerations from mechanism design by introducing hidden types; see also [30], [27], and [14] for more work on information design and Bayesian persuasion.

## 2 Preliminaries

**Model.** We consider a setting with $n$ data buyers, $N = \{1, \ldots, n\}$, each facing a decision problem under uncertainty. The *state of the world*, $\omega$, is unknown and is drawn from a finite state space $\Omega = \{\omega_1, \ldots \omega_m\}$. Each buyer $i$ can choose an *action*, from a finite set $\mathcal{A}_i$. Let $\mathcal{A} = \Pi_{i=1}^n \mathcal{A}_i$. The *ex post* payoff to buyer $i$ for choosing action $a_i \in \mathcal{A}_i$ under state $\omega$ is given by $\mathcal{U}_i(\omega, a_i)$. Unless otherwise specified, we consider the case of matching utility payoffs where the buyer seeks to match

the state and the action [2]. In such settings, we have $|\Omega| = |\mathcal{A}_i|$ for each $i \in [n]$, and the payoff is given by $\mathcal{U}_i(\omega, a_i) = v_i \cdot \mathbf{1}\{\omega = a_i\}$ where each $v_i$ is drawn independently from a distribution $\mathcal{V}_i$. Let $\mathcal{V} = \Pi_{i=1}^n \mathcal{V}_i$.

Each buyer $i$ also has an *interim* belief, $\theta_i \in \Delta(\Omega)$, about the world state. Each buyer's belief, $\theta_i$, is drawn independently from a distribution $\Theta_i$. Let $\Theta = \Pi_{i=1}^n \Theta_i$. The *type* of a buyer is given by the tuple $(v_i, \theta_i)$ and is denoted by $\rho_i$. If the buyers don't vary in $v_i$ or their interim beliefs $\theta_i$, then $\rho_i = \theta_i$ or $\rho_i = v_i$, respectively. Let $\mathcal{P}_i \triangleq \mathcal{V}_i \times \Theta_i$. The utility of a buyer $i \in [n]$ is given by the utility function $u_i : \mathcal{A} \times \Omega \times \mathcal{P}_i \to \mathbb{R}$. We assume that the utility of buyer $i$ depends only on its own type $\rho_i$ but can depend on other players' actions as a negative externality. Additionally, we follow [12] and assume that this externality is *separable*, thus simplifying our utility $u_i(a, \omega, \rho_i) = v_i \cdot \mathbf{1}\{a_i = \omega\} - E_{-i}(a_{-i}, \omega, \rho_i)$. In this paper, we consider settings where the negative externality is given by $E_{-i}(a_{-i}, \omega, \rho_i) = \frac{\alpha}{n-1} \sum_{j \in [n] \setminus i} v_i \cdot \mathbf{1}\{a_j = \omega\}$ where $\alpha \in \mathbb{R}_{\geq 0}$ where $\alpha$ captures the degree of competitiveness among buyers. Let $\bar{\alpha} = \frac{\alpha}{n-1}$

**Statistical Experiments.** There is a data seller who observes the world state and wishes to sell information to one or more buyers to maximize expected revenue. The seller sells information through signaling schemes, where each scheme is called an experiment. The seller chooses, for each buyer $i \in [n]$, a set of signals $\mathcal{S}_i$ and a signaling scheme $\sigma_i : \Omega \to \triangle \mathcal{S}_i$. If state $\omega \in \Omega$ is realized, then the seller sends a signal drawn from the distribution $\sigma_i(\omega)$. Upon receiving a signal, the buyers update their prior beliefs and choose an optimal action accordingly. The signaling scheme $\sigma_i$ can also be represented as a matrix $\pi_i$ — a collection of $m$ row vectors each of dimensions $|\mathcal{S}_i|$. The $j$-th row vector (for $j \in [m]$) specifies the likelihood of each signal when state $\omega_j$ is realized. Thus we have $\sigma_i(\omega_j) = \pi_{i,j}$.

**The Mechanism Design Problem.** The mechanism design goal is to design a set of experiments and corresponding prices to maximize the expected revenue of the seller. Let $\mathcal{B}$ denote a message (bid) space. Define the signaling schemes for a buyer $i \in [n]$ as $\sigma_i : \Omega \times \mathcal{B} \to \triangle \mathcal{S}_i$ and a payment function $t_i : \mathcal{B} \to \mathbb{R}_{\geq 0}$. Given bids $b = (b_1, \ldots b_n) \in \mathcal{B}$, if state $\omega \in \Omega$ is realized, then the seller sends a signal drawn from the distribution $\sigma_i(\omega, b)$ to buyer $i$ and collects payment $t_i(b)$. The sequence of interactions between one or more buyers and the seller takes place as follows:

1. The seller commits to a mechanism, $\mathcal{M} = (\sigma, t)$, where $\sigma = (\sigma_1, \ldots, \sigma_n)$ is a choice and $t = (t_1, \ldots, t_n)$.
2. Each buyer $i$ observes their type, $\rho_i$. The seller observes the state, $\omega$.
3. Each buyer reports a message $b_i$.
4. The seller sends buyer $i$ a signal, $S_i \in \mathcal{S}_i$, generated according to the signaling scheme $\sigma_i(\omega, b)$, and collects payment $t_i(b)$.
5. Each buyer $i$ chooses an action $a_i \in \mathcal{A}_i$, and obtains utility $u_i(a, \omega, \rho_i) - t_i(b)$, where $a = (a_1, \ldots, a_n)$.

By the revelation principle for dynamic games [43, Section 6.3], as long as we consider incentive-compatible mechanisms, it is without loss of generality for the message space to be the type space of buyers, and for the size of the signal space to be the size of the action space, i.e., for each $i \in [n]$, $|\mathcal{S}_i| = |\mathcal{A}_i|$. In such mechanisms, the seller designs experiments where every signal leads to a different optimal choice of action. Following [7], we can then replace every signal as an action recommended by the seller.

**Incentive Compatibility.** A mechanism $(\sigma, t)$ is *Bayesian incentive compatible* (BIC) if the buyer maximizes their expected utility (over other agents' reports) by both reporting their true type as well as by following the recommended actions. For the sake of notational convenience, let $\hat{\mathbb{E}}_{-i} := \mathbb{E}_{\rho_{-i} \sim \mathcal{P}_{-i}}$ denote the operator used for computing the interim representations.

For each $(\rho_i, \rho_i') \in \mathcal{P}_i^2$ and for each deviation function $\delta : \mathcal{A}_i \to \mathcal{A}_i$, a BIC mechanism satisfies:

$$\hat{\mathbb{E}}_{-i} \left[ \mathop{\mathbb{E}}_{\substack{a \sim \sigma(\omega, \rho) \\ \omega \sim \theta_i}} [u_i(a, \omega, \rho) - t_i(\rho)] \right] \geq \hat{\mathbb{E}}_{-i} \left[ \mathop{\mathbb{E}}_{\substack{a \sim \sigma(\omega; \rho_i', \rho_{-i}) \\ \omega \sim \theta_i}} [u_i(\delta(a_i), a_{-i}, \omega, \rho) - t_i(\rho_i', \rho_{-i})] \right] \tag{1}$$

---

[2]It is easier to extend our approach to non-matching utilities as well

In particular, this insists that double deviations (misreporting the type and disobeying the recommendation) are not profitable.

We also consider the stronger notion of *ex post incentive compatible* (IC), which requires, for every agent $i$, and for each $\rho \in \mathcal{P}$, and assuming that every other agent reports its type truthfully and follows the recommended action, then for each misreport $\rho'_i \in \mathcal{P}_i$, and each deviation function, $\delta : \mathcal{A}_i \to \mathcal{A}_i$, the following condition:

$$\mathbb{E}_{\substack{a \sim \sigma(\omega, \rho) \\ \omega \sim \theta_i}} \left[ u_i(a, \omega, \rho) - t_i(\rho) \right] \geq \mathbb{E}_{\substack{a \sim \sigma(\omega; \rho'_i, \rho_{-i}) \\ \omega \sim \theta_i}} \left[ u_i(\delta(a_i), a_{-i}, \omega, \rho) - t_i(\rho'_i, \rho_{-i}) \right] \tag{2}$$

**Individual Rationality.** A mechanism is *interim individually rational* (IIR) if reporting an agent's true type guarantees at least as much expected utility (in expectation over other agents' reports) as opting out of the mechanism. Let $\sigma_{-i} : \Omega \times \mathcal{P}_{-i} \to \triangle \mathcal{A}_{-i}$ denote the recommendations to other participating buyers when buyer $i$ opts out. For each agent $i$, and each $\rho_i \in \mathcal{P}_i$, IIR requires

$$\hat{\mathbb{E}}_{-i} \left[ \mathbb{E}_{\substack{a \sim \sigma(\omega; \rho) \\ \omega \sim \theta_i}} \left[ u_i(a; \omega, \rho) - t_i(\rho) \right] \right] \geq \max_{\tilde{a}_i \in \mathcal{A}_i} \left( \hat{\mathbb{E}}_{-i} \left[ \mathbb{E}_{\substack{a_{-i} \sim \sigma_{-i}(\omega; \rho_{-i}), \\ \omega \sim \theta_i}} \left[ u_i(\tilde{a}_i, a_{-i}; \omega, \rho) \right] \right] \right) \tag{3}$$

It is in the seller's best interest to instantiate the recommendation function to the other buyers when buyer $i$ opts out, $\sigma_{-i}$, to minimize the value of the RHS of this IIR inequality. For this reason, it is without loss of generality to rewrite the above equation as:

$$\hat{\mathbb{E}}_{-i} \left[ \mathbb{E}_{\substack{a \sim \sigma(\omega; \rho) \\ \omega \sim \theta_i}} \left[ u_i(a; \omega, \rho) - t_i(\rho) \right] \right] \geq \min_{\sigma_{-i}} \max_{\tilde{a}_i \in \mathcal{A}_i} \left( \hat{\mathbb{E}}_{-i} \left[ \mathbb{E}_{\substack{a \sim \sigma_{-i}(\omega; \rho_{-i}) \\ \omega \sim \theta_i}} \left[ u_i(\tilde{a}_i, a_{-i}; \omega, \rho) \right] \right] \right) \tag{4}$$

In particular, and recognizing that a buyer's utility decreases the more informed other buyers are, the seller can achieve this by sending optimal recommendations to participating buyers in order to minimize the utility of a non-participating buyer.

We also consider a stronger version of IR, namely *ex post individual rationality* (or simply IR). In this case, for each agent $i$, and each $\rho \in \mathcal{P}$, we require:

$$\mathbb{E}_{\substack{a \sim \sigma(\omega; \rho) \\ \omega \sim \theta_i}} \left[ u_i(a; \omega, \rho) - t_i(\rho) \right] \geq \min_{\sigma_{-i}} \max_{\tilde{a}_i \in \mathcal{A}_i} \left( \mathbb{E}_{\substack{a \sim \sigma_{-i}(\omega; \rho_{-i}), \\ \omega \sim \theta_i}} \left[ u_i(\tilde{a}_i, a_{-i}; \omega, \rho) \right] \right) \tag{5}$$

## 3 Optimal Data Market Design in the Single-Buyer Setting

In this section, we formulate the problem of optimal data market design for a single buyer as an unsupervised learning problem. We study a parametric class of mechanisms, $(\sigma^w, t^w) \in \mathcal{M}$, for parameters, $w \in \mathbb{R}^d$, where $d > 0$. For a single buyer, the goal is to learn parameters $w$ that maximize $\mathbb{E}_{\rho \sim \mathcal{P}} \left[ t^w(\rho) \right]$, such that $(\sigma^w, t^w)$ satisfy *ex post* IC and IR.[3] By adopting differentiable loss functions, we can make use of tools from automatic differentiation and stochastic gradient descent (SGD). For notational convenience, we drop the subscript in this case, as there is only one buyer.

**Neural Network Architecture.** For the single buyer setting, any IC mechanism can be represented as a menu of experiments. For this, we extend the RochetNet architecture to represent a menu of priced statistical experiments. Specifically, the parameters correspond to a menu of $P$ choices, where each choice $p \in [P]$ is associated with an experiment, with parameters $\gamma^p \in \mathbb{R}^{|\Omega| \times |A|}$, and a price, $\beta^p \in \mathbb{R}$. Given this, we define a *menu entry* as,

$$\pi^p_{\omega, j} = \frac{\exp(\gamma^p_{\omega, j})}{\sum_{k \in [m]} \exp(\gamma^p_{\omega, k})}, \quad t^p = \beta^p \tag{6}$$

---

[3]For the single bidder settings, *ex post* IC and IR are the same as BIC and IIR.

The input layer takes $\rho = (v, \theta)$ as input and the network computes the $P$ utility values corresponding to each menu entry. We do not impose obedience constraint explicitly. However, while computing the utility of a menu, we take into consideration the best possible deviating action an agent can take. From the perspective of buyer, if action $k$ is recommended by an experiment choice $p$, then $Pr[\omega = j | a_k, \theta] = \frac{\theta_j \pi_{j,k}^p}{Pr[s_k]}$. The best action deviation is thus $\delta(k) = \arg\max_{j \in [m]} \frac{\theta_j \pi_{j,k}^p}{Pr[s_k]}$, yielding a utility of $v \cdot \max_{j \in [m]} \frac{\theta_j \pi_{j,k}^p}{Pr[s_k]} = \frac{v \cdot \max_{j \in [m]} \theta_j \pi_{j,k}^p}{Pr[s_k]}$. Taking an expectation over all signals, buyer $i$ receives the following utility for the choice $p \in [P]$:

$$h^p(\rho) = v \cdot \left( \sum_{k \in [m]} max_{j \in [m]}\{\theta_j \pi_{j,k}^p\} \right) - \beta^p \tag{7}$$

We also include an additional experiment, $\pi^0$, which corresponds to a *null experiment* that generates the same signal regardless of state, and thus provides no useful information. This menu entry has a price of $t^0 = 0$, and thus $h^0(\rho) = v \cdot (\max_{j \in [m]} \theta_j)$. In particular, this menu entry has the same utility as that of opting out.

**Lemma 3.1.** *Let* $p^*(\rho) = \arg\max_{p \in \{0, \dots P\}} h^p(\rho)$ *denote the optimal menu choice. Then the mechanism* $\mathcal{M} = (\sigma^w, t^w)$ *where* $\sigma^w(\omega, \rho) = \pi_\omega^{p^*(\rho)}$ *and* $t^w(\rho) = t^{p^*(\rho)}$ *satisfies IC and IR.*

This mechanism is IC as it is agent optimizing, and is IR as it guarantees a buyer at least the utility of opting out.

**Training Problem.** The expected revenue of such an IC and IR mechanism that is parameterized by $\gamma, \beta$ is given by $\mathbb{E}_{\rho \sim \mathcal{P}}[\beta^{p^*(\rho)}]$. Rather than minimizing a loss function that measures errors against ground truth labels (as in a supervised learning setting), our goal is to minimize expected negated revenue. To ensure that the objective is differentiable, we replace the argmax operation with a softmax during training. The *loss function*, for parameters $\gamma$ and $\beta$, is thus given by $\mathcal{L}(\gamma, \beta) = \mathbb{E}_{\rho \sim \mathcal{P}}\left[ -\sum_{p=1}^P \beta^p \tilde{\nabla}^p(\rho) \right]$ where $\tilde{\nabla}^p(\rho) = \text{softmax}(\frac{h^1(\rho)}{\tau}, \dots, \frac{h^P(\rho)}{\tau})$, and $\tau > 0$ controls the quality of approximation. Note that, at test time, we revert to using the hard max, so as to guarantee exact IC and IR from a trained network.

During training, this null experiment $(\pi^0, t^0)$ remains fixed, while the parameters $\gamma \in \mathbb{R}^{p \times |\Omega| \times m}$ and $\beta \in \mathbb{R}^p$ are optimized through SGD on the empirical version of the loss calculated over $\ell$ i.i.d samples $\mathcal{S} = \{\rho^{(1)}, \dots \rho^{(\ell)}\}$ drawn from $\mathcal{P}$. We report our results on a separate test set sampled from $\mathcal{P}$. We refer to the Appendix A.2 for more details regarding the hyperparameters.

## 4 Optimal Data Market Design in the Multi-Buyer Setting

In the multi-buyer setting, the goal is to learn parameters $w \in \mathbb{R}^d$ that maximize $\mathbb{E}_{\rho \sim \mathcal{S}}[\sum_{i=1}^n t_w^i(\rho)]$, for a parametric class of mechanisms, $(\sigma^w, t^w) \in \mathcal{M}$, such that $(\sigma^w, t^w)$ satisfy IC (or BIC) and IR (or IIR). By restricting our loss computations to differentiable functions, we can again use tools from automatic differentiation and SGD. For the multi-buyer setting, we use differentiable approximations to represent the rules of the mechanism and compute the degree to which IC constraints are violated during training adopting an augmented Lagrangian method [32].

**Neural Network Architecture.** The neural network architecture has two components: one that encodes the experiments and another that encodes payments. We model these components in a straightforward way as feed-forward, fully connected neural networks with Leaky ReLU activation functions. The input layer consists of the reported type profile, $\rho$, encoded as a vector. For both IC and BIC settings, the component that encodes experiments outputs a matrix $\pi$ of dimensions $n \times |\Omega| \times |\Omega|$, which represents an experiment that corresponds to each of the $n$ agents. In order to ensure feasibility, i.e., the probability values of sending signals for each agent under each state realization is non-negative and sums up to 1, the neural network first computes an output matrix, $\tilde{\pi}$, of the same dimension. For all $i, j, \omega$, we then obtain $\pi_{i,\omega,j}$ by computing $\exp(\tilde{\pi}_{i,\omega,j}) / \sum_k \exp(\tilde{\pi}_{i,\omega,k})$.

For the setting of IC (vs. BIC), we define payments by first computing a *normalized payment*, $\tilde{t}_i^w(\rho) \in [0, 1]$, for each buyer $i$, using a sigmoid activation unit. The payment $t_i^w(\rho)$ is computed as

follows:

$$t_i^w(\rho) = \tilde{t}_i^w(\rho) \cdot v_i \cdot \left( \sum_{k \in [m]} \pi_{i,k,k}^w(\rho)\theta_{i,k} - \bar{\alpha} \sum_{k \in [m]} \sum_{j \in [n] \setminus i} \pi_{j,k,k}^w(\rho)\theta_{j,k} - (\max \theta_i - \alpha) \right) \quad (8)$$

**Lemma 4.1.** *Any mechanism $\mathcal{M} = (\sigma^w, t^w)$ where $\sigma_i^w(\omega, \rho) = \left( \pi_{i,\omega}^w(\rho) \right)$ and $t^w$ satisfies Eqn 8 is ex post IR constraint for any $w \in \mathbb{R}^d$.*

For the BIC setting we only need to compute the *interim* payment, as we can replace $\mathbb{E}_{\rho_{-i} \sim \mathcal{P}_{-i}}[t_i(\rho_i, \rho_{-i})]$ by $t_i(\rho_i)$. For this, we compute an interim normalized payment, $\tilde{t}_i \in [0, 1]$, for each buyer $i$ by using a sigmoid unit. We compute the *interim* payment as:

$$t_i^w(\rho_i) = \tilde{t}_i^w(\rho_i) \cdot v_i \cdot \hat{\mathbb{E}}_{-i} \left[ \sum_{k \in [m]} \pi_{i,k,k}^w(\rho)\theta_{i,k} - \bar{\alpha} \sum_{k \in [m]} \sum_{j \in [n] \setminus i} \pi_{j,k,k}^w(\rho)\theta_{j,k} - (\max \theta_i - \alpha) \right] \quad (9)$$

**Lemma 4.2.** *Any mechanism $\mathcal{M} = (\sigma^w, t^w)$ where $\sigma_i^w(\omega, \rho) = \left( \pi_{i,\omega}^w(\rho) \right)$ and $t^w$ satisfies Eqn 9 is interim IR for any $w \in \mathbb{R}^d$.*

**Training Problem.** In order to train the neural network, we need to minimize the negated revenue subject to incentive constraints. Following Dütting et al. [32], we measure the extent to which a mechanism violates the IC (or BIC) constraints through the notion of *ex post* regret (or interim regret) and then appeal to Lagrangian optimization. The regret for an agent is given by the maximum increase in utility, considering all possible misreports and all possible deviations for a given misreport in consideration while fixing the truthful reports of others (or in expectation over truthful reports of others for the BIC setting) when the others are truthful and obedient.

We define the *ex post* regret for a buyer $i$ as $RGT_i^w = \mathbb{E}_{\rho \in \mathcal{P}} \left[ \max_{\rho_i' \in \mathcal{P}_i} rgt_i^w(\rho_i', \rho) \right]$ where $rgt_i^w(\rho_i', \rho)$ is defined as:

$$rgt_i^w(\rho_i', \rho) = v_i \cdot \left( \sum_{k \in [m]} \max_{k' \in [m]} \left\{ \pi_{i,k',k}^w(\rho_i', \rho_{-i})\theta_{i,k'} \right\} - \bar{\alpha} \sum_{k \in [m]} \sum_{j \in [n] \setminus i} \pi_{j,k,k}^w(\rho_i', \rho_{-i})\theta_{j,k} \right)$$

$$- v_i \cdot \left( \sum_{k \in [m]} \pi_{i,k,k}^w(\rho)\theta_{i,k} - \bar{\alpha} \sum_{k \in [m]} \sum_{j \in [n] \setminus i} \pi_{j,k,k}^w(\rho)\theta_{j,k} \right) - (t_i^w(\rho_i', \rho_{-i}) - t_i^w(\rho)) \quad (10)$$

**Lemma 4.3.** *Any mechanism $\mathcal{M} = (\sigma^w, t^w)$ where $\sigma_i^w(\omega, \rho) = \left( \pi_{i,\omega}^w(\rho) \right)$ is ex post IC if and only if $RGT_i^w = 0 \; \forall i \in [n]$, except for measure zero events.*

We define the *interim* regret for a buyer $i$ as $\widehat{RGT}_i^w = \mathbb{E}_{\rho \in \mathcal{P}} \left[ \max_{\rho_i' \in \mathcal{P}_i} \widehat{rgt}_i^w(\rho_i', \rho_i) \right]$ where $\widehat{rgt}_i^w(\rho_i', \rho_i)$ is defined as:

$$\widehat{rgt}_i^w(\rho_i', \rho_i) = v_i \cdot \left( \sum_{k \in [m]} \max_{k' \in [m]} \hat{\mathbb{E}}_{-i} \left[ \pi_{i,k',k}^w(\rho_i', \rho_{-i})\theta_{i,k'} - \bar{\alpha} \sum_{j \in [n] \setminus i} \pi_{j,k,k}^w(\rho_i', \rho_{-i})\theta_{j,k} \right] \right)$$

$$- v_i \cdot \hat{\mathbb{E}}_{-i} \left[ \sum_{k \in [m]} \pi_{i,k,k}^w(\rho_i, \rho_{-i})\theta_{i,k} - \bar{\alpha} \sum_{k \in [m]} \sum_{j \in [n] \setminus i} \pi_{j,k,k}^w(\rho_i, \rho_{-i})\theta_{j,k} \right] - (t_i(\rho_i') - t_i(\rho_i))$$

$$(11)$$

**Lemma 4.4.** *Any mechanism $\mathcal{M} = (\sigma^w, t^w)$ where $\sigma_i^w(\omega, \rho) = \left( \pi_{i,\omega}^w(\rho) \right)$ is interim IC if and only if $\widehat{RGT}_i^w = 0 \; \forall i \in [n]$, except for measure zero events.*

Given regret, we compute the Lagrangian objective on the empirical version of loss and regret over samples $\mathcal{S} = \{\rho^{(1)}, \ldots, \rho^{(L)}\}$. We solve this objective using augmented Lagrangian optimization.

The BIC setting also involves an inner expectation to compute the interim representations. There, for each agent $i$, we sample a separate subset $\mathcal{S}_{-i} = \{\rho_{-i}^{(1)}, \ldots, \rho_{-i}^{(K)}\}$, and replace the inner expectations with an empirical expectation over these samples. We sample several misreports, compute the best performing misreport among these samples, and use this as a warm-start initialization for the inner maximization problem. For the BIC setting, rather than sampling fresh misreports and computing the interim experiments and payments, we re-use the other samples from the minibatch and their already computed interim values to find the best initialization. This leads to a dramatic speed-up as the number of forward passes required to compute the regret reduces by an order magnitude of the number of initialized misreports. This, however, works only for the BIC setting, as the constraints are only dependent on the interim values.

We report all our results on a separate test set sampled from $\mathcal{P}$. Please refer to Appendix B.6 for more details regarding the hyperparameters.

## 5 Experimental Results for the Single Buyer Setting

In this section, we demonstrate the use of RochetNet to recover known existing results from the economic theory literature. Additionally, we also show how we can use this approach to conjecture the optimal menu structure for settings outside the reach of current theory. Further, we give in the Appendix C.2 representative examples of how we can characterize the differential informativeness of the menu options when we change different properties of the prior distribution. This builds economic intuition in regard to the shape of optimal market designs.

**Buyers with world prior heterogeneity.** We first show that RochetNet can recover the optimal menu for all the continuous distribution settings considered in [7] where the input distributions are a continuum of types. We specifically consider the following settings with binary states and binary actions with payoffs $v = 1$ and interim beliefs drawn from:

A. an unit interval, i.e, $\theta \sim U[0, 1]$.

B. an equal weight mixture of $Beta(8, 30)$ and $Beta(60, 30)$.

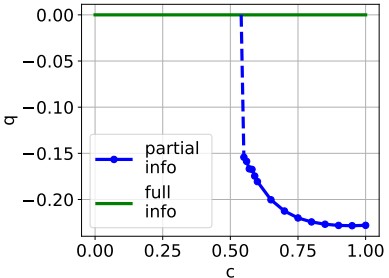

Figure 1: The differential informativeness $q$ of the menu(s) learned by RochetNet for Setting D, varying the parameter $c$ that instantiates the economic environment.

The optimal menus for each of these settings are given by [7]. For the first setting, the optimal menu consists of a fully informative experiment with a price of $0.25$. For the second setting, the seller offers two menu options: a partially informative experiment and a fully informative experiment. In each case, RochetNet recovers the exact optimal menu. We describe the optimal menus, associated prices, revenue, and RochetNet revenue in the Appendix C.1.

We also give the results from additional experiments in Appendix C.2, where we demonstrate how the informativeness of the menu $q$ changes, as we vary different properties of the economic environment.

**Buyers with payoff and world prior heterogeneity.** We are aware of no theoretical characterization of optimal data market designs when both $v$ and $\theta$ vary. In such cases, we can use RochetNet to conjecture the structure of an optimal solution. For this, we consider the following settings with enlarged buyer types with binary states and binary actions with:

C. $v \sim U[0, 1]$ and the interim beliefs are drawn from $U[0, 1]$.

D. $v \sim U[c, 1]$ and the interim beliefs are drawn from an equal weight mixture of $Beta(8, 30)$ and $Beta(60, 30)$ for $c \in [0, 1]$.

For Setting C, RochetNet learns a menu consisting of a single fully informative experiment with a price of $0.14$. For Setting D, RochetNet learns a menu consisting of a single fully informative experiment when $c < 0.55$. For higher values of $c$, RochetNet learns an additional partially informative menu option. In Figure 1, we show how the differential informativeness $q$ of the partially informative menu changes with $c$.

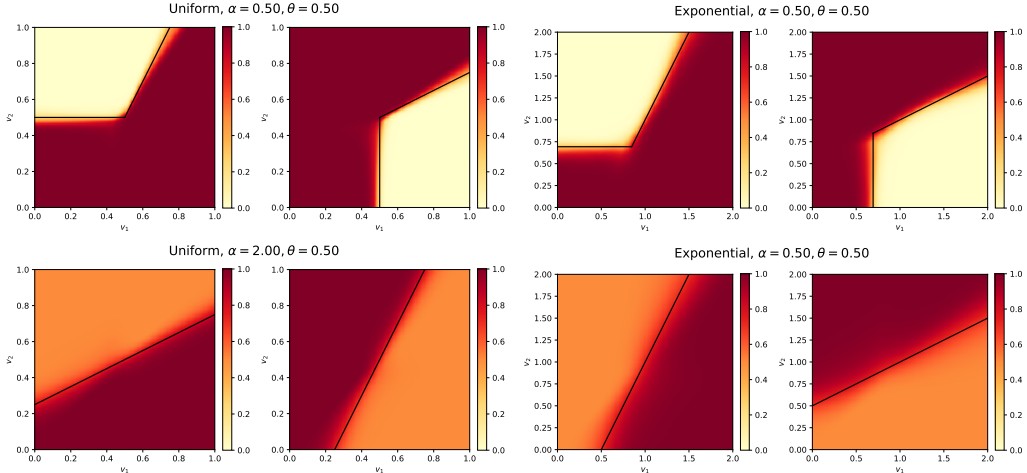

Figure 2: Experiments learned for Settings E and F, for BIC constraints (**top**) and *ex post* IC constraints (**bottom**). The plot shows the probability of recommending the correct action to buyer 1 (left subplot) and buyer 2 (right subplot) for varying values of $v_1$ (x-axis) and $v_2$ (y-axis). For each setting, the theoretically optimal experiments are represented by solid lines separating the regions.

## 6   Experimental Results for the Multi-Buyer Setting

In this section, we show how we can use RegretNet to recover known existing results for the multi-buyer, BIC setting. We also use RegretNet to conjecture the optimal solution for a multi-buyer problem in the *ex post* IC setting, and then prove the optimality of the design. Lastly, we give results for a setting where it is analytically hard to compute the optimal solution but RegretNet is used to understand how revenue varies with changes to the intensity of negative externality, again building economic intuition for the market design problem.

**BIC settings.**   [12] study the multi-buyer market design problem with two buyers, binary actions, and fixed interim beliefs, the same for each agent. They show that a deterministic signaling scheme is optimal, and thus characterize their solution in terms of recommending a correct action (selling a fully informative menu). Since the obedience constraints are defined on the interim representation, the optimal mechanism is also able to sometimes recommend incorrect actions to a buyer (in effect sending recommendations that are opposed with the realized state).

We consider the following settings from [12] with interim beliefs are given by $\theta_1 = \theta_2 = (0.5, 0.5)$ and $\alpha = 0.5$ and:

  E.  The payoffs $v_1, v_2$ are sampled from the unit interval $U[0, 1]$.

  F.  The payoffs $v_1, v_2$ are sampled from the exponential distribution with $\lambda = 1$.

The results for these settings with a BIC constraint are shown in Figure 2 (top). The theoretically optimum solution contains two kinds of recommendations, those recommending the correct action (the optimal action to take given the realized state), and those recommending incorrect actions, separated by the black solid lines. The heatmap showed in the plots denotes our computational results, which show the probability of recommending the correct action. In particular, we can confirm visually (and from its expected revenue) that RegretNet is able to recover the optimal revenue as well the optimal experiment design. The test revenue and regret along with additional results are in Appendix D.1.

*Ex post* **IC settings.**   We apply RegretNet to the same three settings, but now adopting *ex post* IC constraints. This is of interest because there is no known analytical solution for the optimal mechanism for two buyers in the *ex post* IC setting. See Figure 2 (bottom). Based on this, and additional experiments (in Appendix D.2 ), we were able to conjecture that the structure of the optimal design in this ex post IC setting, and we prove its optimality by following Myerson's framework for single item auctions. We defer the proof to the appendix D.4.

**Theorem 6.1.** *Consider the setting with Binary State and Binary Actions where the buyers have a common interim belief $\theta_i = \theta$. The payoff $v_i$ for a buyer $i$ is drawn from a regular distribution $\mathcal{V}_i$ with a continuous density function. Define virtual value $\phi_i(v_i) = v_i - \frac{1-F(v_i)}{f(v_i)}$, for pdf $f$ and cdf $F$ of distribution $\mathcal{V}_i$. The revenue-optimal mechanism satisfying* ex post *IC and IR is a mechanism that sells the fully informative experiment to buyer $i$ if $\phi_i(v_i) \geq \bar{\alpha} \sum_{i \neq j} \phi_j(v_j)$. Otherwise buyer $i$ receives an uninformative menu, where the signal corresponds to the most likely state based on the prior.*

**Studying the effect of varying the intensity of externalities.** Following [12], we study the effect of the negative externality parameter $\alpha$ on the revenue for the BIC, multi-buyer setting with a common and known prior for each buyer and payoffs sampled from $U[0,1]$. We also consider the case where the payoffs are constant but the interim beliefs are independently sampled from $U[0,1]$. While there is no known analytical characterization for the latter, we show how easy it is to study the models learnt by RegretNet to analyze economic properties of interest.

Figure 3 shows the effect of $\alpha$ on the revenue. In the context of uncertain priors, we note that the effect on revenue is similar to the setting where payoffs are uncertain. As we enhance competition intensity via $\alpha$, the revenue grows. Even though the buyers are not recommended the correct action, the seller manages to generate revenue by threatening to share exclusive information with a competitor. This effect becomes fiercer in the settings with uncertain priors.

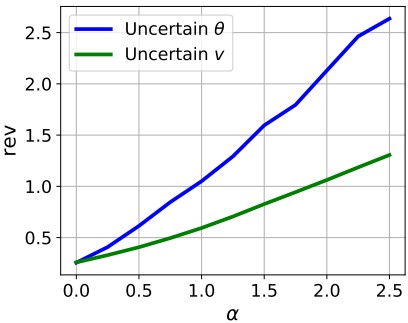

Figure 3: Effect of $\alpha$ on revenue

## 7 Conclusion

We have introduced a new deep neural network architecture and learning framework to study the design of optimal data markets. We have demonstrated through experimental work the flexibility of the framework, showing that it can adapt to relatively complex scenarios and facilitate the discovery of new optimal designs. Note that while we have only considered matching utility payoffs in this paper, our approach can easily be extended to non-matching utility payoffs as well.

We also point out some limitations. First, for our approach to continue to provide insights into the theoretically optimal design for larger problems (e.g., with more buyers, more signals, more actions), it will be important to provide interpretability to the mechanisms learned by RegretNet (designs learned by RochetNet, on the other hand, are immediately interpretable). Second, while our approach scales well with the number of buyers or states in the *ex post IC* setting, it does not scale as easily with the number of buyers in the BIC setting. The challenge in the BIC setting comes from the interim computations involving conditional expectations over reports of others and scaling beyond what is studied in this paper will require new techniques, for example, exploiting symmetry. Third, we are making use of gradient-based approaches, which may suffer from local optima in non-convex problems. At the same time, deep learning has shown success in various problem domains despite non-convexity. The experiments reported here align with these observations, with our neural network architectures consistently recovering optimal solutions, when these are known, and thus optimality can be verified, and despite non-convex formulations. Fourth, we attain in the multi-buyer setting only approximate, and not exact, incentive alignment, and this leaves the question of how much alignment is enough for agents to follow the intended advice of a market design (there is little practical or theoretical guidance in this regard). Moreover, we do not know for a given solution whether there is an exact IC solution nearby. While there is some recent guiding theory [22, 26, 15] that provides transformations between $\epsilon$-BIC and BIC without revenue loss in the context of auction design, extending these transformations to approximate IC settings and to problems with both types and actions presents an interesting avenue for future research.

Lastly, we return to where we started, and underline that markets for trading data about individuals raise substantive ethical concerns [1, 2, 5, 8, 11, 19, 33]. Our hope is that machine learning frameworks such as those introduced here can be used to strike new kinds of trade-offs, for example allowing individuals to also benefit directly from trades on data about themselves and to embrace privacy and fairness constraints.

## Acknowledgements

We would like to thank Zhe Feng and Alessadro Bonatti for the initial discussions. The source code for all experiments is available from Github at `https://github.com/saisrivatsan/deep-data-markets/`

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

# A Data Markets in the Single-Buyer Settings

## A.1 RochetNet Architecture

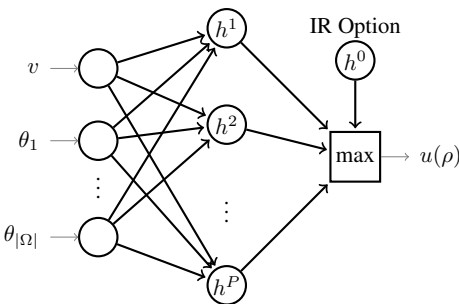

Figure 4: RochetNet: Neural network architecture for learning the optimal menu for a single bidder setting. The neural network takes in as inputs the buyer's payoff $v$ and interim belief $\theta$ (a distribution on world state) and computes the utility to the buyer for each of $P + 1$ menu options, where option 0 corresponds to the uninformative experiment.

The neural network architecture for the single buyer settings is depicted in Fig 4. In this architecture, each menu entry $p$ is represented by a matrix $\pi^p$ with dimensions $|\Omega| \times |\Omega|$, along with an associated price $t^p$. Instead of explicitly enforcing the obedience constraint, we consider all potential actions an agent may take to deviate from the recommended action to compute the utility. This utility for the menu option $p$ is given by Eqn 7 in the main paper.

## A.2 Setup and Hyper-parameters

**Training:** We train RochetNet with $P = 1000$ possible menu entries. We set the softmax temperature $\tau$ to $\frac{1}{200}$. We train RochetNet for $20,000$ iterations with a minibatch of size $2^{15}$ sampled online for every update.

We report all our results on a test-set of 20000 samples that are separate from the train set.

**Time:** For the settings studied in the paper, RochetNet take 7-8 minutes minutes to train on a single NVIDIA Tesla V100 GPU.

# B Data Markets in the Multi-Buyer Setting

## B.1 RegretNet Architecture

The neural network architecture for the multi-buyer *ex post* IC setting is depicted in Fig 5. For the BIC setting, the experiment network $\sigma$ remains the same and the interim experiments are computed through sampling. They payment network only takes the type corresponding to the buyer as input as we can replace $\tilde{\mathbb{E}}_{-i}[t_i(\rho_i, \rho_{-i})]$ with just $t_i(\rho_i)$. We also have separate payment networks for each buyer if their type distributions are different.

## B.2 Proof of Lemma 4.1

**Lemma 4.1.** *Any mechanism $\mathcal{M} = (\sigma^w, t^w)$ where $\sigma_i^w(\omega, \rho) = (\pi_{i,\omega}^w(\rho))$ and $t^w$ satisfies Eqn 8 is ex post IR constraint for any $w \in \mathbb{R}^d$.*

*Proof.* For the *ex post* IC setting, we first compute a *normalized payment*, $\tilde{t}_i^w(\rho) \in [0, 1]$, for each buyer $i$ by using a sigmoidal unit. We scale this by the difference between utility achieved by the buyer when he is truthful and obedient and the utility of opting out. Thus, the payment $t_i^w(\rho)$ is

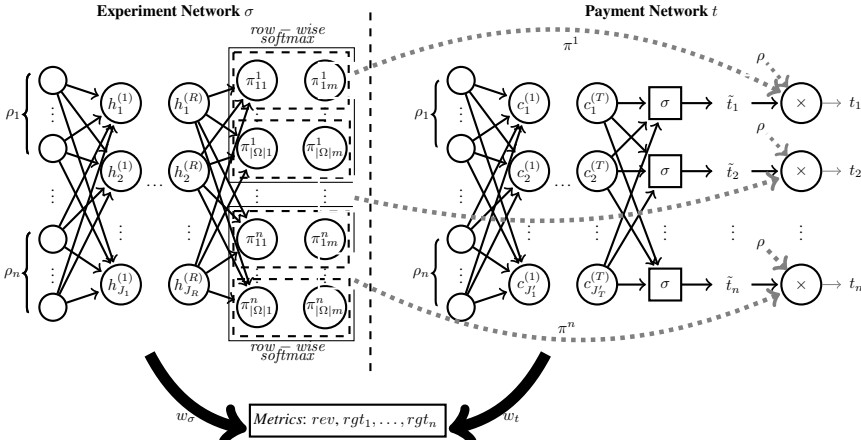

**Figure 5:** RegretNet: Neural network architecture for the multi-buyer *ex post* IC setting. The inputs are the reported types $\rho$ of each buyer. The experiment network outputs an experiment of size $|\Omega| \times |\Omega|$ for each buyer, where a row-wise softmax operation ensures the experiments are well defined. The payment network outputs the normalized payment $\tilde{t}_i^w(\rho) \in [0,1]$ through a sigmoid activation unit that is used to compute payment via Equation 8.

computed as follows:

$$t_i^w(\rho) = \tilde{t}_i^w(\rho) \cdot v_i \cdot \left( \sum_{k \in [m]} \pi_{i,k,k}^w(\rho)\theta_{i,k} - \bar{\alpha} \sum_{k \in [m]} \sum_{j \in [n]\setminus i} \pi_{j,k,k}^w(\rho)\theta_{j,k} - (\max_k \theta_{i,k} - \alpha) \right) \quad (12)$$

To establish ex post IR, we reason about the utility a buyer can obtain when not participating in the mechanism. As per the discussion in Section 2, in an optimal data market the seller will send optimal recommendations to participating buyers in the event that buyer $i$ opts out, thus minimizing the utility of the buyer who opts out. In particular, the recommendation for any $j \neq i$, for an opting out buyer $i$, is such that $\mathbb{E}_{\substack{a \sim \sigma_{-i}(\omega;\rho_{-i}) \\ \omega \sim \theta_i}}[\mathbf{1}\{a_j = \omega\}] = 1$. Given this, the utility of buyer $i$, when opting out, is

$$\min_{\sigma_{-i}} \max_{\tilde{a}_i \in \mathcal{A}_i} \left( \mathop{\mathbb{E}}_{\substack{a \sim \sigma_{-i}(\omega;\rho_{-i}), \\ \omega \sim \theta_i}} [u_i(\tilde{a}_i, a_{-i}; \omega, \rho)] \right)$$

$$= \max_{\tilde{a}_i \in \mathcal{A}_i} \left( \mathbb{E}_{\substack{a \sim \sigma_{-i}(\omega;\rho_{-i}) \\ \omega \sim \theta_i}} \left[ v_i \cdot \mathbf{1}\{\tilde{a}_i = \omega\} - v_i\bar{\alpha} \sum_{j \in [n]\setminus i} \mathbf{1}\{\tilde{a}_i = \omega\} \right] \right) \quad (13)$$

$$= \max_{\tilde{a}_i \in \mathcal{A}_i} \left( \mathbb{E}_{\omega \sim \theta_i} [v_i \cdot \mathbf{1}\{\tilde{a}_i = \omega\} - v_i\alpha] \right)$$

$$= v_i \cdot (\max \theta_i - \alpha)$$

Note that $\sum_{k \in [m]} \pi_{i,k,k}^w(\rho)\theta_{i,k} \geq \max_k \theta_{i,k}$ and $\bar{\alpha} \left( \sum_{k \in [m]} \sum_{j \in [n]\setminus i} \pi_{j,k,k}^w(\rho)\theta_{j,k} \right) \leq \bar{\alpha} \left( \sum_{j \in [n]\setminus i} 1 \right) = \alpha$. Thus we have $\sum_{k \in [m]} \pi_{i,k,k}^w(\rho)\theta_{i,k} - \bar{\alpha} \left( \sum_{k \in [m]} \sum_{j \in [n]\setminus i} \pi_{j,k,k}^w(\rho)\theta_{j,k} \right) - (\max_k \theta_{i,k} - \alpha) \geq 0$.

Taking this with the fact that $\tilde{t}_i^w(\rho) \in [0,1]$, we have

$$t_i^w(\rho) = \tilde{t}_i^w(\rho) \cdot v_i \cdot \left( \sum_{k \in [m]} \pi_{i,k,k}^w(\rho)\theta_{i,k} - \bar{\alpha} \sum_{k \in [m]} \sum_{j \in [n] \setminus i} \pi_{j,k,k}^w(\rho)\theta_{j,k} - \left( \max_k \theta_{i,k} - \alpha \right) \right)$$

$$\leq v_i \cdot \left( \mathop{\mathbb{E}}_{\substack{a \sim \sigma_i^w(\omega;\rho), \\ \omega \sim \theta_i}} \left[ \left( \mathbf{1}\{a_i = \omega\} - \bar{\alpha} \sum_{j \in [n] \setminus i} \mathbf{1}\{a_j = \omega\} \right) \right] - \left( \max_k \theta_{i,k} - \alpha \right) \right) \tag{14}$$

$$= \mathop{\mathbb{E}}_{\substack{a \sim \sigma_i^w(\omega;\rho), \\ \omega \sim \theta_i}} \left[ v_i \cdot \left( \mathbf{1}\{a_i = \omega\} - \bar{\alpha} \sum_{j \in [n] \setminus i} \mathbf{1}\{a_j = \omega\} \right) \right] - v_i \cdot \left( \max_k \theta_{i,k} - \alpha \right)$$

$$= \mathop{\mathbb{E}}_{\substack{a \sim \sigma_i^w(\omega;\rho), \\ \omega \sim \theta_i}} [u_i(a;\omega,\rho)] - v_i \cdot (\max \theta_i - \alpha))$$

Rearranging, and substituting $v_i \cdot (\max \theta_i - \alpha)$ from Eqn 13, we get the IR constraint described in Eqn 5. $\qquad \square$

### B.3  Proof of Lemma 4.2

**Lemma 4.2.** *Any mechanism $\mathcal{M} = (\sigma^w, t^w)$ where $\sigma_i^w(\omega, \rho) = \left( \pi_{i,\omega}^w(\rho) \right)$ and $t^w$ satisfies Eqn 9 is interim IR for any $w \in \mathbb{R}^d$.*

*Proof.* For the BIC setting, we work with the *interim* payment, and make use of a normalized interim payment, $\tilde{t}_i(\rho_i) \in [0,1]$, for each buyer $i$, by using a sigmoidal unit. In this case, we scale this normalized payment by the difference between the interim utility achieved when the buyer is truthful and obedient and the interim utility of opting out. Thus, the *interim* payment is

$$t_i^w(\rho_i) = \tilde{t}_i^w(\rho_i) \cdot v_i \cdot \hat{\mathbb{E}}_{-i} \left[ \sum_{k \in [m]} \pi_{i,k,k}^w(\rho)\theta_{i,k} - \bar{\alpha} \sum_{k \in [m]} \sum_{j \in [n] \setminus i} \pi_{j,k,k}^w(\rho)\theta_{j,k} - (\max \theta_i - \alpha) \right] \tag{15}$$

The utility of the outside option is again computed similarly to the *ex post* IC setting, recognizing that the seller in the optimal mechanism will make recommendation $\hat{\mathbb{E}}_{-i} \left[ \mathbb{E}_{\substack{a \sim \sigma_{-i}(\omega;\rho_{-i}), \\ \omega \sim \theta_i}} [\mathbf{1}\{a_j = \omega\}] \right] = 1$ to any agent $j \neq i$, in the case that $i$ drops out. In particular, the utility of buyer $i$, when opting out, is

$$\min_{\sigma_{-i}} \max_{\tilde{a}_i \in \mathcal{A}_i} \left( \hat{\mathbb{E}}_{-i} \left[ \mathbb{E}_{\substack{a \sim \sigma_{-i}(\omega;\rho_{-i}), \\ \omega \sim \theta_i}} [u_i(\tilde{a}_i, a_{-i}; \omega, \rho)] \right] \right)$$

$$= \max_{\tilde{a}_i \in \mathcal{A}_i} \left( \hat{\mathbb{E}}_{-i} \left[ \mathbb{E}_{\substack{a \sim \sigma_{-i}(\omega;\rho_{-i}), \\ \omega \sim \theta_i}} \left[ v_i \cdot \mathbf{1}\{\tilde{a}_i = \omega\} - v_i \bar{\alpha} \sum_{j \in [n] \setminus i} \mathbf{1}\{a_j = \omega\} \right] \right] \right) \tag{16}$$

$$= \max_{\tilde{a}_i \in \mathcal{A}_i} \left( \mathbb{E}_{\omega \sim \theta_i} [v_i \cdot \mathbf{1}\{\tilde{a}_i = \omega\} - v_i \alpha] \right)$$

$$= v_i \cdot (\max \theta_i - \alpha)$$

Note that $\hat{\mathbb{E}}_{-i} \left[ \sum_{k \in [m]} \pi_{i,k,k}^w(\rho)\theta_{i,k} \right] \geq \max_k \theta_{i,k}$ and $\hat{\mathbb{E}}_{-i} \left[ \bar{\alpha} \left( \sum_{k \in [m]} \sum_{j \in [n] \setminus i} \pi_{j,k,k}^w(\rho)\theta_{j,k} \right) \right] \leq \bar{\alpha} \left( \sum_{j \in [n] \setminus i} 1 \right) = \alpha$. Thus we have

$$\hat{\mathbb{E}}_{-i} \left[ \sum_{k \in [m]} \pi_{i,k,k}^w(\rho)\theta_{i,k} - \bar{\alpha} \sum_{k \in [m]} \sum_{j \in [n] \setminus i} \pi_{j,k,k}^w(\rho)\theta_{j,k} - \left( \max_k \theta_{i,k} - \alpha \right) \right] \geq 0$$

Taking this with the fact that $\tilde{t}_i^w(\rho_i) \in [0,1]$, we have

$$
\begin{aligned}
t_i^w(\rho_i) &= \tilde{t}_i^w(\rho_i) \cdot v_i \cdot \hat{\mathbb{E}}_{-i}\left[\sum_{k \in [m]} \pi_{i,k,k}^w(\rho)\theta_{i,k} - \bar{\alpha} \sum_{k \in [m]}\sum_{j \in [n]\setminus i} \pi_{j,k,k}^w(\rho)\theta_{j,k} - (\max \theta_i - \alpha)\right] \\
&\leq v_i \cdot \hat{\mathbb{E}}_{-i}\left[\sum_{k \in [m]} \pi_{i,k,k}^w(\rho)\theta_{i,k} - \bar{\alpha} \sum_{k \in [m]}\sum_{j \in [n]\setminus i} \pi_{j,k,k}^w(\rho)\theta_{j,k}\right] - v_i \cdot (\max \theta_i - \alpha)) \\
&= \hat{\mathbb{E}}_{-i}\left[\mathop{\mathbb{E}}_{\substack{a \sim \sigma_i^w(\omega;\rho), \\ \omega \sim \theta_i}}\left[v_i \cdot \left(\mathbf{1}\{a_i = \omega\} - \bar{\alpha}\sum_{j \in [n]\setminus i}\mathbf{1}\{a_j = \omega\}\right)\right]\right] - v_i \cdot (\max \theta_i - \alpha)) \\
&\leq \hat{\mathbb{E}}_{-i}\left[\mathop{\mathbb{E}}_{\substack{a \sim \sigma_i^w(\omega;\rho), \\ \omega \sim \theta_i}}[u_i(a;\omega,\rho)]\right] - v_i \cdot (\max \theta_i - \alpha))
\end{aligned}
$$

(17)

Rearranging the above and substituting $v_i \cdot (\max \theta_i - \alpha)$ from Eqn 16, we get the IIR constraint described in Eqn 4.

$\square$

## B.4 Proof of Lemma 4.3

**Lemma 4.3.** *Any mechanism* $\mathcal{M} = (\sigma^w, t^w)$ *where* $\sigma_i^w(\omega,\rho) = \left(\pi_{i,\omega}^w(\rho)\right)$ *is ex post IC if and only if* $RGT_i^w = 0 \ \forall i \in [n]$, *except for measure zero events.*

*Proof.* We first prove the forward direction: if any mechanism $\mathcal{M} = (\sigma^w, t^w)$ where $\sigma_i^w(\omega,\rho) = \left(\pi_{i,\omega}^w(\rho)\right)$ is ex post IC, then $RGT_i^w = 0, \forall i \in [n]$.

For the *ex post* IC setting, the incentive compatibility constraints requires that for every agent $i$, and for each $\rho \in \mathcal{P}$, and assuming that every other agent reports its type truthfully and follows the recommended action, then for each misreport $\rho_i' \in \mathcal{P}_i$, and each deviation function, $\delta : \mathcal{A}_i \to \mathcal{A}_i$, the following condition:

$$
\mathop{\mathbb{E}}_{\substack{a \sim \sigma(\omega,\rho), \\ \omega \sim \theta_i}}[u_i(a,\omega,\rho) - t_i(\rho)] \geq \mathop{\mathbb{E}}_{\substack{a \sim \sigma(\omega;\rho_i',\rho_{-i}), \\ \omega \sim \theta_i}}[u_i(\delta(a_i), a_{-i}, \omega, \rho) - t_i(\rho_i', \rho_{-i})]
$$

(18)

Note that the above hold for any deviation function $\delta$. Therefore, we can write that

$$
\mathop{\mathbb{E}}_{\substack{a \sim \sigma(\omega,\rho), \\ \omega \sim \theta_i}}[u_i(a,\omega,\rho) - t_i(\rho)] \geq \max_\delta \mathop{\mathbb{E}}_{\substack{a \sim \sigma(\omega;\rho_i',\rho_{-i}), \\ \omega \sim \theta_i}}[u_i(\delta(a_i), a_{-i}, \omega, \rho) - t_i(\rho_i', \rho_{-i})]
$$

(19)

We can expand to get the following equation:

$$
\begin{aligned}
&v_i \cdot \left(\sum_{k \in [m]} \pi_{i,k,k}^w(\rho)\theta_{i,k} - \bar{\alpha}\sum_{k \in [m]}\sum_{j \in [n]\setminus i}\pi_{j,k,k}^w(\rho)\theta_{j,k}\right) - t_i(\rho) \\
&\geq \max_\delta v_i \cdot \left(\sum_{k \in [m]}\left\{\pi_{i,\delta(k),k}^w(\rho_i',\rho_{-i})\theta_{i,\delta(k)}\right\} - \bar{\alpha}\sum_{k \in [m]}\sum_{j \in [n]\setminus i}\pi_{j,k,k}^w(\rho_i',\rho_{-i})\theta_{j,k}\right) - t_i(\rho_i', \rho_{-i})
\end{aligned}
$$

(20)

Pushing the max inside (since other terms don't involve $\delta(k)$), we note that:

$$
\max_\delta \sum_{k \in [m]}\left\{\pi_{i,\delta(k),k}^w(\rho_i',\rho_{-i})\theta_{i,\delta(k)}\right\} = \sum_{k \in [m]}\max_{k'}\left\{\pi_{i,k',k}^w(\rho_i',\rho_{-i})\theta_{i,k'}\right\}
$$

(21)

Thus we have,

$$v_i \cdot \left( \sum_{k \in [m]} \pi_{i,k,k}^w(\rho)\theta_{i,k} - \bar{\alpha} \sum_{k \in [m]} \sum_{j \in [n]\setminus i} \pi_{j,k,k}^w(\rho)\theta_{j,k} \right) - t_i(\rho)$$

$$\geq v_i \cdot \left( \sum_{k \in [m]} \max_{k' \in [m]} \{\pi_{i,k',k}^w(\rho'_i, \rho_{-i})\theta_{i,k'}\} - \bar{\alpha} \sum_{k \in [m]} \sum_{j \in [n]\setminus i} \pi_{j,k,k}^w(\rho'_i, \rho_{-i})\theta_{j,k} \right) - t_i(\rho'_i, \rho_{-i}) \tag{22}$$

Note that by our definition of *ex post* regret in Equation 10, the above is equivalent to $rgt_i^w(\rho'_i, \rho) \leq 0$. Furthermore, this holds for any deviating report $\rho'_i$. Thus, we can write:

$$\max_{\rho'_i \in \mathcal{P}_i} rgt_i^w(\rho'_i, \rho) \leq 0 \tag{23}$$

But note that

$$\max_{\rho'_i \in \mathcal{P}_i} rgt_i^w(\rho'_i, \rho) \geq rgt_i^w(\rho_i, \rho)$$

$$= v_i \cdot \left( \sum_{k \in [m]} \max_{k' \in [m]} \{\pi_{i,k',k}^w(\rho)\theta_{i,k'}\} - \bar{\alpha} \sum_{k \in [m]} \sum_{j \in [n]\setminus i} \pi_{j,k,k}^w(\rho)\theta_{j,k} \right)$$

$$- v_i \cdot \left( \sum_{k \in [m]} \pi_{i,k,k}^w(\rho)\theta_{i,k} - \bar{\alpha} \sum_{k \in [m]} \sum_{j \in [n]\setminus i} \pi_{j,k,k}^w(\rho)\theta_{j,k} \right) \tag{24}$$

$$= v_i \cdot \left( \sum_{k \in [m]} \left[ \max_{k' \in [m]} \{\pi_{i,k',k}^w(\rho)\theta_{i,k'}\} - \pi_{i,k,k}^w(\rho)\theta_{i,k} \right] \right)$$

$$\geq 0$$

where the last step holds since $\max_{k' \in [m]} \left\{ \pi_{i,k',k}^w(\rho)\theta_{i,k'} \right\} \geq \pi_{i,k,k}^w(\rho)\theta_{i,k}$

Combining Equations 23 and 24, we have $\max_{\rho'_i \in \mathcal{P}_i} rgt_i^w(\rho'_i, \rho) = 0$. Taking the expectation over all profiles $\rho \in \mathcal{P}$, we have $\mathbb{E}_{\rho \in \mathcal{P}} \left[ \max_{\rho'_i \in \mathcal{P}_i} rgt_i^w(\rho'_i, \rho) \right] = 0$. Thus, $RGT_i^w = 0, \forall i \in [n]$. Thus, we've shown that if any mechanism $\mathcal{M} = (\sigma^w, t^w)$ where $\sigma_i^w(\omega, \rho) = \left( \pi_{i,\omega}^w(\rho) \right)$ is *ex post* IC, then $RGT_i^w = 0$, as desired.

Next, we consider the reverse direction: if $RGT_i^w = 0, \forall i \in [n]$, we want to show that any mechanism $\mathcal{M} = (\sigma^w, t^w)$ where $\sigma_i^w(\omega, \rho) = \left( \pi_{i,\omega}^w(\rho) \right)$ is ex post IC. Starting with $RGT_i^w = 0, \forall i \in [n]$, we have that by definition:

$$\mathbb{E}_{\rho \in \mathcal{P}} \left[ \max_{\rho'_i \in \mathcal{P}_i} rgt_i^w(\rho'_i, \rho) \right] = 0 \tag{25}$$

We showed in Equation 24 that $rgt_i^w(\rho_i, \rho) \geq 0$. Since $\rho_i \in \mathcal{P}_i$, we have that $\max_{\rho'_i \in \mathcal{P}_i} rgt_i^w(\rho'_i, \rho) \geq 0$. Together with the fact that $\mathbb{E}_{\rho \in \mathcal{P}} \left[ \max_{\rho'_i \in \mathcal{P}_i} rgt_i^w(\rho'_i, \rho) \right] = 0$, we have that $\max_{\rho'_i \in \mathcal{P}_i} rgt_i^w(\rho'_i, \rho) = 0 \implies rgt_i^w(\rho'_i, \rho) \leq 0$.

Plugging in our definition for *ex post* regret, we have that:

$$v_i \cdot \left( \sum_{k \in [m]} \max_{k' \in [m]} \{\pi_{i,k',k}^w(\rho'_i, \rho_{-i})\theta_{i,k'}\} - \bar{\alpha} \sum_{k \in [m]} \sum_{j \in [n]\setminus i} \pi_{j,k,k}^w(\rho'_i, \rho_{-i})\theta_{j,k} \right)$$

$$- v_i \cdot \left( \sum_{k \in [m]} \pi_{i,k,k}^w(\rho)\theta_{i,k} - \bar{\alpha} \sum_{k \in [m]} \sum_{j \in [n]\setminus i} \pi_{j,k,k}^w(\rho)\theta_{j,k} \right) - (t_i^w(\rho'_i, \rho_{-i}) - t_i^w(\rho)) \leq 0 \tag{26}$$

This can be transformed back to:

$$\mathbb{E}_{\substack{a\sim\sigma(\omega,\rho)\\\omega\sim\theta_i}}[u_i(a,\omega,\rho)-t_i(\rho)] \geq \max_\delta \mathbb{E}_{\substack{a\sim\sigma(\omega;\rho_i',\rho_{-i}),\\\omega\sim\theta_i}}[u_i(\delta(a_i),a_{-i},\omega,\rho)-t_i(\rho_i',\rho_{-i})]$$

$$\geq \mathbb{E}_{\substack{a\sim\sigma(\omega;\rho_i',\rho_{-i})\\\omega\sim\theta_i}}[u_i(\delta(a_i),a_{-i},\omega,\rho)-t_i(\rho_i',\rho_{-i})] \tag{27}$$

This is exactly the *ex post* IC constraint described in Equation 2

$\square$

## B.5 Proof of Lemma 4.4

**Lemma 4.4.** *Any mechanism* $\mathcal{M} = (\sigma^w, t^w)$ *where* $\sigma_i^w(\omega,\rho) = \left(\pi_{i,\omega}^w(\rho)\right)$ *is interim IC if and only if* $\widehat{RGT}_i^w = 0 \; \forall i \in [n]$, *except for measure zero events.*

*Proof.* We first prove the forward direction: if any mechanism $\mathcal{M} = (\sigma^w, t^w)$ where $\sigma_i^w(\omega,\rho) = \left(\pi_{i,\omega}^w(\rho)\right)$ is interim IC, then $\widehat{RGT}_i^w = 0, \forall i \in [n]$.

For the BIC setting, for each $(\rho_i, \rho_i') \in \mathcal{P}_i^2$ and for each deviation function $\delta : \mathcal{A}_i \to \mathcal{A}_i$, a BIC mechanism satisfies:

$$\hat{\mathbb{E}}_{-i}\left[\mathbb{E}_{\substack{a\sim\sigma(\omega,\rho)\\\omega\sim\theta_i}}[u_i(a,\omega,\rho)-t_i(\rho)]\right] \geq \hat{\mathbb{E}}_{-i}\left[\mathbb{E}_{\substack{a\sim\sigma(\omega;\rho_i',\rho_{-i})\\\omega\sim\theta_i}}[u_i(\delta(a_i),a_{-i},\omega,\rho)-t_i(\rho_i',\rho_{-i})]\right] \tag{28}$$

Note that the above hold for any deviation function $\delta$, therefore we can write that

$$\hat{\mathbb{E}}_{-i}\left[\mathbb{E}_{\substack{a\sim\sigma(\omega,\rho)\\\omega\sim\theta_i}}[u_i(a,\omega,\rho)-t_i(\rho)]\right] \geq \max_\delta \hat{\mathbb{E}}_{-i}\left[\mathbb{E}_{\substack{a\sim\sigma(\omega;\rho_i',\rho_{-i})\\\omega\sim\theta_i}}[u_i(\delta(a_i),a_{-i},\omega,\rho)-t_i(\rho_i',\rho_{-i})]\right] \tag{29}$$

We can expand to get the following equations:

$$v_i \cdot \hat{\mathbb{E}}_{-i}\left[\sum_{k\in[m]}\pi_{i,k,k}^w(\rho_i,\rho_{-i})\theta_{i,k} - \bar\alpha\sum_{k\in[m]}\sum_{j\in[n]\backslash i}\pi_{j,k,k}^w(\rho_i,\rho_{-i})\theta_{j,k}\right] - t_i(\rho_i)$$

$$\geq \max_\delta v_i \cdot \hat{\mathbb{E}}_{-i}\left[\left(\sum_{k\in[m]}\pi_{i,\delta(k),k}^w(\rho_i',\rho_{-i})\theta_{i,\delta(k)} - \bar\alpha\sum_{k\in[m]}\sum_{j\in[n]\backslash i}\pi_{j,k,k}^w(\rho_i',\rho_{-i})\theta_{j,k}\right)\right] - t_i(\rho_i')$$

$$= \max_\delta v_i \cdot \left(\sum_{k\in[m]}\hat{\mathbb{E}}_{-i}\left[\pi_{i,\delta(k),k}^w(\rho_i',\rho_{-i})\theta_{i,\delta(k)} - \bar\alpha\sum_{j\in[n]\backslash i}\pi_{j,k,k}^w(\rho_i',\rho_{-i})\theta_{j,k}\right]\right) - t_i(\rho_i')$$

$$= v_i \cdot \left(\sum_{k\in[m]}\max_\delta \hat{\mathbb{E}}_{-i}\left[\pi_{i,\delta(k),k}^w(\rho_i',\rho_{-i})\theta_{i,\delta(k)} - \bar\alpha\sum_{j\in[n]\backslash i}\pi_{j,k,k}^w(\rho_i',\rho_{-i})\theta_{j,k}\right]\right) - t_i(\rho_i')$$

$$= v_i \cdot \left(\sum_{k\in[m]}\max_{k'\in[m]} \hat{\mathbb{E}}_{-i}\left[\pi_{i,k',k}^w(\rho_i',\rho_{-i})\theta_{i,\delta(k)} - \bar\alpha\sum_{j\in[n]\backslash i}\pi_{j,k,k}^w(\rho_i',\rho_{-i})\theta_{j,k}\right]\right) - t_i(\rho_i') \tag{30}$$

The above equations follow from the linearity of expectations and from pushing the max to the inside (since other terms don't involve $\delta(k)$).

Note that by our definition of *interim* regret in Equation 11, the above equation is equivalent to $\widehat{rgt}_i^w(\rho_i',\rho_i) \leq 0$. Furthermore, this holds for any deviating report $\rho_i'$. Therefore, we can write that

$$\max_{\rho_i' \in \mathcal{P}_i} \widehat{rgt}_i^w (\rho_i', \rho_i) \leq 0 \tag{31}$$

But note that

$$\max_{\rho_i' \in \mathcal{P}_i} \widehat{rgt}_i^w (\rho_i', \rho) \geq \widehat{rgt}_i^w (\rho_i, \rho_i)$$

$$= v_i \cdot \left( \sum_{k \in [m]} \max_{k' \in [m]} \hat{\mathbb{E}}_{-i} \left[ \pi_{i,k',k}^w(\rho_i, \rho_{-i})\theta_{i,k'} - \bar{\alpha} \sum_{j \in [n] \backslash i} \pi_{j,k,k}^w(\rho_i, \rho_{-i})\theta_{j,k} \right] \right) - t_i(\rho_i)$$

$$- v_i \cdot \hat{\mathbb{E}}_{-i} \left[ \sum_{k \in [m]} \pi_{i,k,k}^w(\rho_i, \rho_{-i})\theta_{i,k} - \bar{\alpha} \sum_{k \in [m]} \sum_{j \in [n] \backslash i} \pi_{j,k,k}^w(\rho_i, \rho_{-i})\theta_{j,k} \right] + t_i(\rho_i)$$

$$= v_i \cdot \left( \sum_{k \in [m]} \max_{k' \in [m]} \hat{\mathbb{E}}_{-i} \left[ \pi_{i,k',k}^w(\rho_i, \rho_{-i})\theta_{i,k'} \right] - \hat{\mathbb{E}}_{-i} \left[ \pi_{i,k,k}^w(\rho_i, \rho_{-i})\theta_{i,k} \right] \right)$$

$$\geq 0 \tag{32}$$

where the second step follows from linearity of expectation, and the last step holds since we are taking the max of all possible $k' \in [m]$, which includes $k' = k$.

Combining Equations 31 and 32, we have that $\max_{\rho_i' \in \mathcal{P}_i} \widehat{rgt}_i^w (\rho_i', \rho_i) = 0$. Taking the expectation over all profiles $\rho$, we have $\hat{\mathbb{E}}_{\rho \in \mathcal{P}} \left[ \max_{\rho_i' \in \mathcal{P}_i} \widehat{rgt}_i^w (\rho_i', \rho_i) \right] = 0 \implies \widehat{RGT}_i^w = 0, \forall i \in [n]$. Thus, we've showed that if any mechanism $\mathcal{M} = (\sigma^w, t^w)$ where $\sigma_i^w(\omega, \rho) = \left( \pi_{i,\omega}^w(\rho) \right)$ is BIC, then $\widehat{RGT}_i^w = 0$, as desired.

Next, we consider the reverse direction: if $\widehat{RGT}_i^w = 0, \forall i \in [n]$, we want to show that any mechanism $\mathcal{M} = (\sigma^w, t^w)$ where $\sigma_i^w(\omega, \rho) = \left( \pi_{i,\omega}^w(\rho) \right)$ is BIC. Starting with $\widehat{RGT}_i^w = 0, \forall i \in [n]$, we have that by definition: $\hat{\mathbb{E}}_{\rho \in \mathcal{P}} \left[ \max_{\rho_i' \in \mathcal{P}_i} \widehat{rgt}_i^w (\rho_i', \rho_i) \right] = 0$

We showed in Equation 32 that $\widehat{rgt}_i^w (\rho_i, \rho_i) \geq 0$. Since $\rho_i \in \mathcal{P}_i$, we have that $\max_{\rho_i' \in \mathcal{P}_i} \widehat{rgt}_i^w (\rho_i', \rho_i) \geq 0$. Together with the fact that $\hat{\mathbb{E}}_{\rho \in \mathcal{P}} \left[ \max_{\rho_i' \in \mathcal{P}_i} \widehat{rgt}_i^w (\rho_i', \rho_i) \right] = 0$, we have that $\max_{\rho_i' \in \mathcal{P}_i} \widehat{rgt}_i^w (\rho_i', \rho_i) = 0 \implies \widehat{rgt}_i^w (\rho_i', \rho_i) \leq 0$. Plugging in our definition for *interim regret*, we have that:

$$v_i \cdot \left( \sum_{k \in [m]} \max_{k' \in [m]} \hat{\mathbb{E}}_{-i} \left[ \pi_{i,k',k}^w(\rho_i', \rho_{-i})\theta_{i,k'} - \bar{\alpha} \sum_{j \in [n] \backslash i} \pi_{j,k,k}^w(\rho_i', \rho_{-i})\theta_{j,k} \right] \right) - t_i(\rho_i')$$

$$\leq v_i \cdot \hat{\mathbb{E}}_{-i} \left[ \sum_{k \in [m]} \pi_{i,k,k}^w(\rho_i, \rho_{-i})\theta_{i,k} - \bar{\alpha} \sum_{k \in [m]} \sum_{j \in [n] \backslash i} \pi_{j,k,k}^w(\rho_i, \rho_{-i})\theta_{j,k} \right] - t_i(\rho_i) \tag{33}$$

This can be transformed back to:

$$\hat{\mathbb{E}}_{-i} \left[ \mathbb{E}_{\substack{a \sim \sigma(\omega, \rho) \\ \omega \sim \theta_i}} [u_i(a, \omega, \rho) - t_i(\rho)] \right] \geq \max_{\delta} \hat{\mathbb{E}}_{-i} \left[ \mathbb{E}_{\substack{a \sim \sigma(\omega; \rho_i', \rho_{-i}) \\ \omega \sim \theta_i}} [u_i(\delta(a_i), a_{-i}, \omega, \rho) - t_i(\rho_i', \rho_{-i})] \right]$$

$$\geq \hat{\mathbb{E}}_{-i} \left[ \mathbb{E}_{\substack{a \sim \sigma(\omega; \rho_i', \rho_{-i}) \\ \omega \sim \theta_i}} [u_i(\delta(a_i), a_{-i}, \omega, \rho) - t_i(\rho_i', \rho_{-i})] \right] \tag{34}$$

This is exactly the *interim* IC constraint described in Equation 1

$\square$

### B.6 Setup and Hyper-parameters

**Training:** For the multi-buyer setting, all our neural networks consist of 3 hidden layers with 200 hidden units each. For the IC setting, we sample a minibatch of 1024 samples online to update the parameters of the neural networks. We sample 100 misreports for every sample to compute the best initialization. For the BIC setting, we use a minibatch of 128 samples. We compute the interim values wherever required over 512 samples drawn from $\mathcal{P}_{-i}$.

We observe that sampling alone is sufficient to approximate the regret while training and do not perform any additional gradient descent steps, as this does not improve the training performance (however we anticipate this step would be necessary for larger settings).

We train the neural networks for 20000 iterations and make parameter updates using the Adam Optimizer with a learning rate of 0.001. The Lagrangian parameters $(\lambda_1, \ldots \lambda_n)$ are set to 10.0. The coefficient of the penalty term is initialized to 1.0. The Lagrangian updates are performed once every 100 iteration.

**Testing:** We report all our results on a test size of 20000 samples. For the *ex post* setting, we use 100 misreports as initialization to warm-start the inner maximization to compute regret. For the BIC setting, we use other samples of the minibatch to compute a defeating misreport as noted in the previous subsection. We then run 100 steps of gradient ascent with a learning rate of 0.005 to compute the defeating misreport and the regret more accurately.

**Training time:** RegretNet takes 11 - 12 minutes per experiment for training for all the settings studied in this paper. All our experiments were run on a single NVIDIA Tesla V100 GPU.

## C Experimental Results for the Single Buyer Setting

### C.1 Buyers with world prior heterogeneity

We present optimal menus, associated prices, revenue, and RochetNet revenue for the Single Buyer Setting A and B in Figure 6

| Distribution | Rochet Menu | | Rochet | Opt |
| | Experiment | Price | $rev$ | $rev$ |
| --- | --- | --- | --- | --- |
| Setting A | $\begin{bmatrix} 1.00 & 0.00 \\ 0.00 & 1.00 \end{bmatrix}$ | 0.25 | 0.125 | 0.125 |
| Setting B | $\begin{bmatrix} 0.78 & 0.22 \\ 0 & 1.00 \end{bmatrix}$ | 0.14 | 0.167 | 0.166 |
| | $\begin{bmatrix} 1.00 & 0.00 \\ 0.00 & 1.00 \end{bmatrix}$ | 0.26 | | |

Figure 6: Menu(s) and associated prices learned by RochetNet, RochetNet revenue and Optimal revenue on test data in Settings A and B. RochetNet recovers the optimal menu's for both these settings

### C.2 Additional Experiments for Buyers with world prior heterogeneity

We design additional experiments where we use RochetNet to study how the informativeness of learned experiments vary when we change different properties of the prior distributions. For this, we consider the mixture of $Beta$ distributions considered in setting B. This is a bimodal distribution with modes at $\theta_L = \left(\frac{7}{36}, \frac{29}{36}\right)$ and $\theta_H = \left(\frac{59}{88}, \frac{29}{88}\right)$. We call the former "low type" and the latter "high type" following the analysis of [7] for the case of binary types, and order this way because

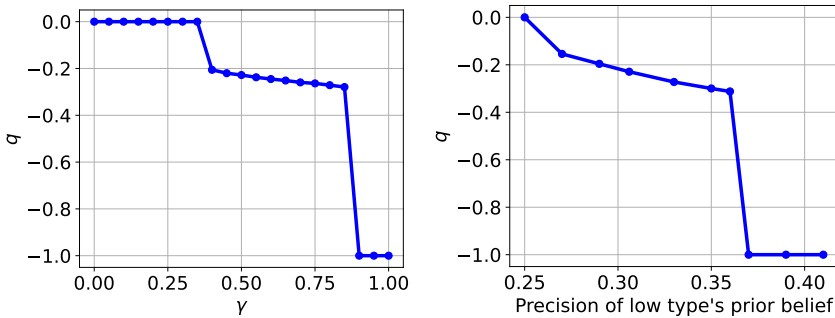

Figure 7: Results for Experiment G (Left) and H (Right). For both experiments, the vertical axis is the differential informativeness $q$ of the experiment used by $\theta_L$. For Experiment G (left), the horizontal axis is the frequency of the high type, and for Experiment H (right), the horizontal axis is the precision of the low type's prior belief $|\theta_L - 0.5|$.

$|\theta_L - 0.5| \geq |\theta_H - 0.5|$. We also call the value ($|\theta - 0.5|$) the precision of a type's prior belief. We design two experiments:

G. Single buyer, Binary State, and Binary Actions, where the payoffs $v_1 = 1$ and the interim beliefs are drawn from a mixture of $Beta(8, 30)$ and $Beta(60, 30)$ weighted by $(1 - \gamma, \gamma)$ for $\gamma \in [0, 1]$.

H. Single buyer, Binary State, and Binary Actions, where the payoffs $v_1 = 1$ and the interim beliefs are drawn from an equal weight mixture of $Beta(a_1, b_1)$ and $Beta(60, 30)$. We vary $a_1, b_1$ so that the mode of $Beta(a_1, b_1)$ decreases (therefore the precision of the low type's prior belief $|\theta_L - 0.5|$ increases) while ensuring that the ratio of values at the modes of these distributions stays constant.

In particular, the modes of the two distributions are $\theta_L = \left( \frac{a-1}{a+b-2}, \frac{b-1}{a+b-2} \right)$ and $\theta_H = \left( \frac{59}{88}, \frac{29}{88} \right)$ in the case if Experiment G.[4]. $\theta_L$ and $\theta_H$ are non-congruent types, i.e., without the supplemental information, a buyer with type $\theta_L$ takes action 0 while the other takes action 1. $\theta_H$ also values a fully informative experiment more than $\theta_L$. In the first experiment, we change the likelihood of type $\theta_H$ with respect to type $\theta_L$. In the second experiment, we change the precision of the belief of one type while keeping the other fixed (we vary values of $a, b$ while numerically ensuring that the probability distribution function of all the plotted points have the same height at the mode).

The results are given in Figure 7. For Experiment G, increasing the high type's frequency decreases the low type experiment's informativeness. For Experiment H, increasing the precision of the low type's prior belief decreases the low type experiment's informativeness. [7] characterize the informativeness of the optimal experiment for similar settings, but for discrete distributions with two types. In particular, Proposition 3 from [7] states that informativeness of the low type $\pi_{1,1}$, decreases when the frequency of the high type or with the precision of low type's prior belief $|\theta_L - 0.5|$. It is interesting, then, that we observe the same behaviour in the RochetNet results for a continuous analog of their discrete distribution set-up. This suggests a new target for economic theory.

## D  Experimental Results for the Multi Buyer Setting

We consider the following multi-buyer data market design problem with two buyers, binary action and fixed interim beliefs, adopting all possible combinations of the following choices in regard to the configuration of the economy:

- $\theta = (0.5, 0.5)$ or $\theta = (0.75, 0.25)$

---

[4]This also requires $|\frac{a-1}{a+b-2} - 0.5| \geq |\frac{59}{88} - 0.5|$, otherwise $Beta(a, b)$ becomes the high type instead. For our experiments, we only consider data points at which this condition holds, in other words, we only consider combinations of $a, b$ such that $Beta(a, b)$ is the low type with mode at $\theta_L$ and $Beta(60, 30)$ is the high type with mode at $\theta_H$.

- $\alpha = 0.5$ or $\alpha = 2.0$
- Both payoffs drawn from UNF, EXP, or Asym UNF

UNF is the uniform distribution over the unit interval $U[0, 1]$. EXP is the exponential distribution with $\lambda = 1$. For the Asym UNF, the payoff for buyer $i$ is uniform over the interval $[0, i + 1]$. In the next two subsections, we present our results for the BIC and *ex post* IC settings, respectively.

## D.1 BIC Settings

Figures 8, 9, and 10 show the optimal data market design and the data marker learned by RegretNet for the BIC settings described above, and Figure 11 gives the test revenue and regret obtained by RegretNet and the revenue of the optimal mechanism.

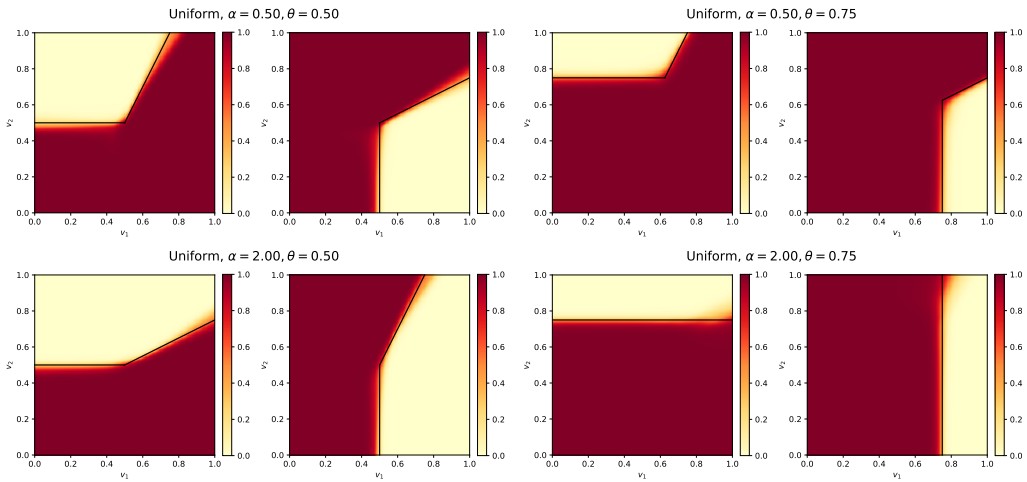

Figure 8: Experiments learned for BIC constraints when the payoffs are drawn from UNF

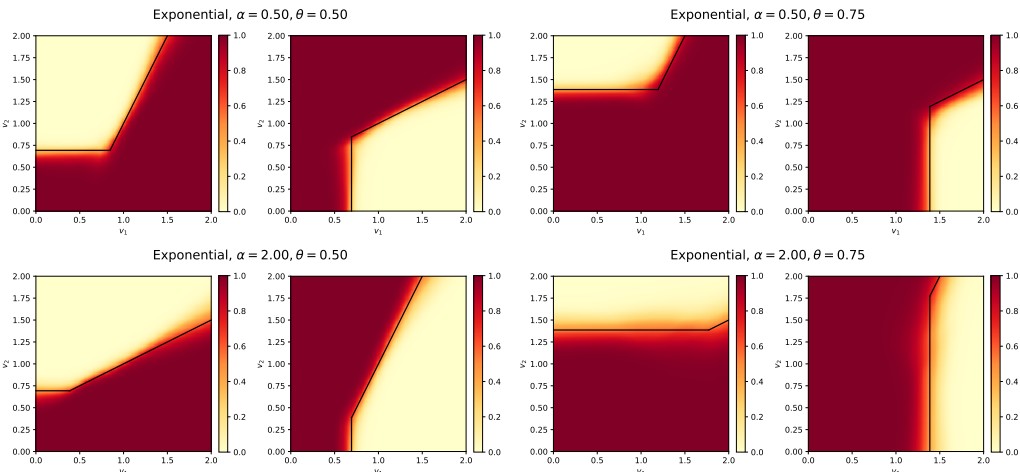

Figure 9: Experiments learned for BIC constraints when the payoffs are drawn from EXP

## D.2 *Ex post* IC Settings

Figures 12, 13, and 14 show the optimal data market design and the data market learned by RegretNet for the *ex post* IC settings described above, and Figure 15 gives the test revenue and test regret obtained by RegretNet along with the revenue of the optimal mechanism.

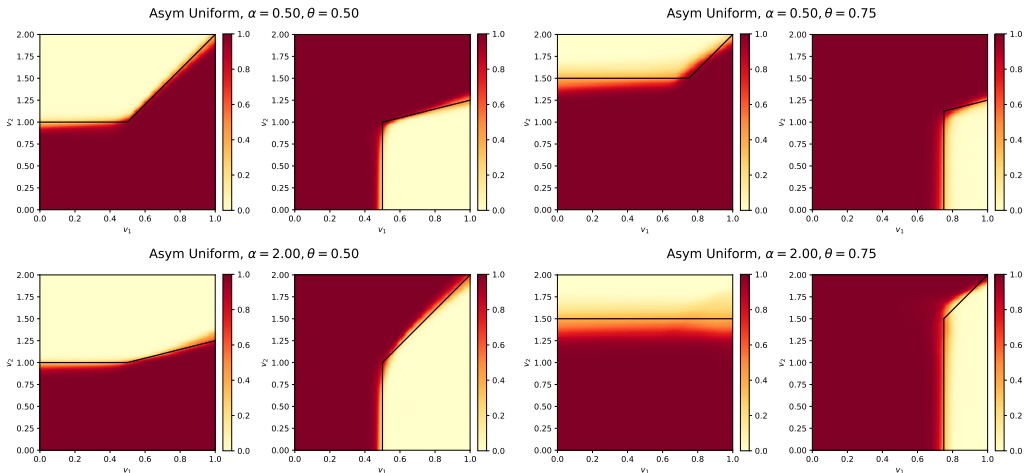

Figure 10: Experiments learned for BIC constraints when the payoffs are drawn from Asym UNF

| Distribution | $\alpha$ | $\theta$ | RegretNet | | Opt |
| --- | --- | --- | --- | --- | --- |
| | | | *rev* | *rgt* | *rev* |
| UNF | 0.5 | (0.5, 0.5) | 0.400 | <0.001 | 0.395 |
| | | (0.75, 0.25) | 0.237 | | 0.237 |
| | 2.0 | (0.5, 0.5) | 1.056 | | 1.042 |
| | | (0.75, 0.25) | 0.763 | | 0.75 |
| EXP | 0.5 | (0.5, 0.5) | 0.632 | <0.001 | 0.632 |
| | | (0.75, 0.25) | 0.448 | | 0.447 |
| | 2.0 | (0.5, 0.5) | 1.603 | | 1.613 |
| | | (0.75, 0.25) | 1.390 | | 1.415 |
| Asym UNF | 0.5 | (0.5, 0.5) | 0.614 | <0.001 | 0.609 |
| | | (0.75, 0.25) | 0.366 | | 0.369 |
| | 2.0 | (0.5, 0.5) | 1.599 | | 1.594 |
| | | (0.75, 0.25) | 1.139 | | 1.135 |

Figure 11: Test Revenue and Test Regret obtained by RegretNet for the BIC Settings.

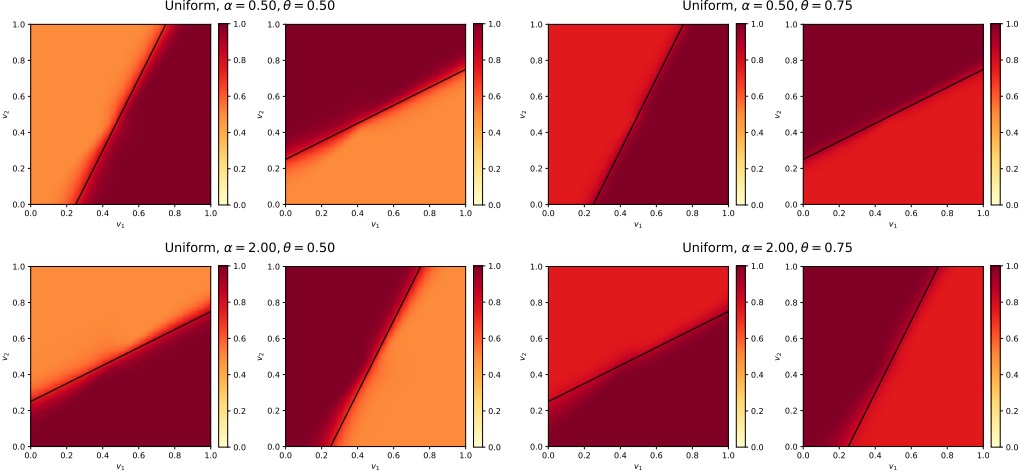

Figure 12: Experiments learned for *ex post* IC constraints when the payoffs are drawn from UNF

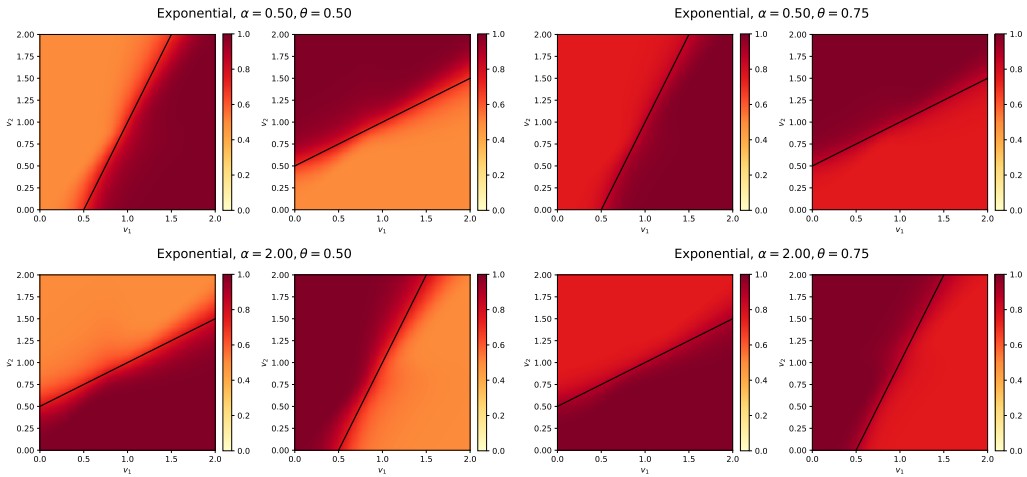

Figure 13: Experiments learned for *ex post* IC constraints when the payoffs are drawn from EXP

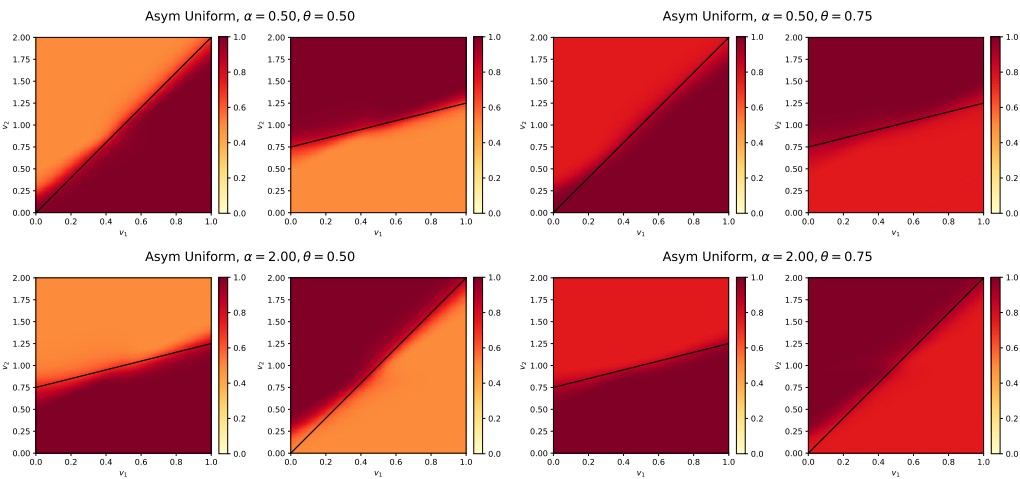

Figure 14: Experiments learned for *ex post* IC constraints when the payoffs are drawn from Asym UNF

| Distribution | $\alpha$ | $\theta$ | RegretNet | | Opt |
| | | | *rev* | *rgt* | *rev* |
|---|---|---|---|---|---|
| UNF | 0.5 | (0.5, 0.5) | 0.277 | <0.001 | 0.27 |
| | | (0.75, 0.25) | 0.14 | | 0.135 |
| | 2.0 | (0.5, 0.5) | 0.553 | | 0.541 |
| | | (0.75, 0.25) | 0.278 | | 0.270 |
| EXP | 0.5 | (0.5, 0.5) | 0.405 | <0.001 | 0.405 |
| | | (0.75, 0.25) | 0.204 | | 0.202 |
| | 2.0 | (0.5, 0.5) | 0.801 | | 0.809 |
| | | (0.75, 0.25) | 0.418 | | 0.405 |
| Asym UNF | 0.5 | (0.5, 0.5) | 0.426 | <0.001 | 0.423 |
| | | (0.75, 0.25) | 0.421 | | 0.21 |
| | 2.0 | (0.5, 0.5) | 0.84 | | 0.841 |
| | | (0.75, 0.25) | 0.423 | | 0.42 |

Figure 15: Test Revenue and Test Regret for RegretNet for the *ex post* IC Settings.

### D.3 Additional Results for the Irregular Distributions

We present additional results for both BIC and *ex post* IC settings for two buyers, binary actions, and interim beliefs $\theta = (0.5, 0.5)$, the same for each buyer. We set $\alpha = 0.5$, and the payoffs are drawn from the irregular distribution whose pdf $f(v)$ is:

$$f(v) = \begin{cases} 2.5 & \text{if } 0 \le v < 0.3 \\ 0.5 & 0.3 \le v < 0.8 \end{cases} \tag{35}$$

The optimal solutions for these problems make use of ironed virtual values as the payoff distribution is irregular (see Fig 16).

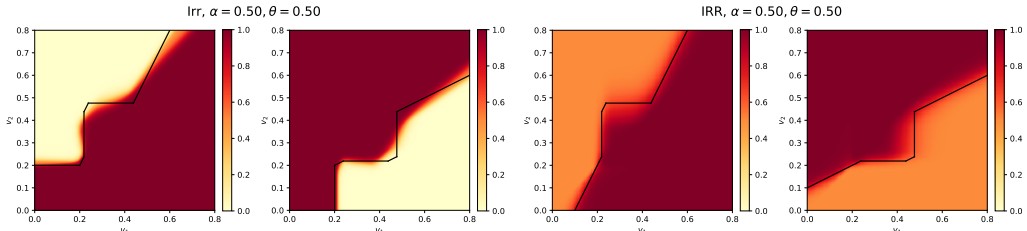

Figure 16: Experiments learned for BIC and *ex post* IC constraints when the payoffs are drawn from an irregular distribution whose pdf is given by Eqn 35. For both these settings, we use the ironed virtual value functions to compute the optimal data market design.

### D.4 Proof of Theorem 6.1

**Theorem 6.1.** *Consider the setting with Binary State and Binary Actions where the buyers have a common interim belief $\theta_i = \theta$. The payoff $v_i$ for a buyer $i$ is drawn from a regular distribution $\mathcal{V}_i$ with a continuous density function. Define virtual value $\phi_i(v_i) = v_i - \frac{1-F(v_i)}{f(v_i)}$, for pdf $f$ and cdf $F$ of distribution $\mathcal{V}_i$. The revenue-optimal mechanism satisfying ex post IC and IR is a mechanism that sells the fully informative experiment to buyer $i$ if $\phi_i(v_i) \ge \bar{\alpha} \sum_{i \ne j} \phi_j(v_j)$. Otherwise buyer $i$ receives an uninformative menu, where the signal corresponds to the most likely state based on the prior.*

*Proof.* We consider a setting with binary states, binary actions, and where each buyer has the same interim belief, set to $(\theta, 1-\theta)$. Without loss of generality, let $\theta \ge (1-\theta)$. Since the interim beliefs are fixed, we will just represent buyer types with $v_i$ instead of $\rho_i$. Let $x_i(v) = \mathbb{E}_{\substack{a \sim \sigma(\omega; \rho), \\ \omega \sim \theta_i}} [\mathbf{1}\{a_i = \omega\}]$.

Let $\pi_i(v)$ be the matrix representation of the experiment assigned to buyer $i$ i.e. $\pi_{i,\omega}(v) = \sigma_i(\omega, v)$

We first show that a mechanism is obedient if and only if the experiment assigned to the buyer satisfies $x_i(v) \ge \theta$. If a mechanism is obedient, then $\theta \cdot \pi_{i,1,1}(v) \ge (1-\theta)\pi_{i,2,1}(v)$ and $(1-\theta)\pi_{i,2,2}(v) \ge \theta \cdot \pi_{i,1,2}(v)$. We have $x_i(v) = \theta \cdot \pi_{i,1,1}(v) + (1-\theta)\pi_{i,2,2}(v) \ge \theta \cdot \pi_{i,1,1}(v) + \theta \cdot \pi_{i,1,2}(v) = \theta(\pi_{i,1,1}(v) + \pi_{i,1,2}(v)) = \theta$.

In the other direction, consider when $x_i(v) \ge \theta$. In this case, we show that both $\theta \cdot \pi_{i,1,1}(v) \ge (1-\theta)\pi_{i,2,1}(v)$ and $(1-\theta)\pi_{i,2,2}(v) \ge \theta \cdot \pi_{i,1,2}(v)$ need to hold for obedience. Assume to the contrary, one of these fails to hold when $x_i(v) \ge \theta$. We have one of,

- $(1-\theta)\pi_{i,2,2}(v) < \theta \cdot \pi_{i,1,2}(v)$, and we have $x_i(v) = \theta \cdot \pi_{i,1,1}(v) + (1-\theta)\pi_{i,2,2}(v) < \theta \cdot \pi_{i,1,1}(v) + \theta \cdot \pi_{i,1,2}(v) = \theta(\pi_{i,1,1}(v) + \pi_{i,1,2}(v)) = \theta$.

- $\theta \cdot \pi_{i,1,1}(v) < (1-\theta)\pi_{i,2,1}(v)$, and we have $x_i(v) = \theta \cdot \pi_{i,1,1}(v) + (1-\theta)\pi_{i,2,2}(v) < (1-\theta)\pi_{i,2,1}(v) + (1-\theta)\pi_{i,2,2}(v) = (1-\theta)(\pi_{i,2,1}(v) + \pi_{i,2,2}(v)) = (1-\theta) \le \theta$.

In either case, we have a contradiction.

Since our payoff is linear, from Proposition 3.5 in [7], the IC constraints are satisfied only when the *truthfulness* and *obedience* constraints are satisfied. Denote $\tilde{x}_i(v) = x_i(v) - \frac{\alpha}{n-1} \sum_{j \in [n] \setminus i} x_j(v)$.

Thus, the optimal design problem to solve is:

$$\mathbb{E}_{v \sim \mathcal{V}} \left[ \sum_{i \in [n]} t_i(v_i) \right]$$

$$\text{s.t. } v_i \tilde{x}_i(v) - t_i(v) \geq v_i \tilde{x}_i(v_i'; v_{-i}) - t_i(v_i'; v_{-i}) \qquad \forall v \in \mathcal{V}, v_i' \in \mathcal{V}_i, i \in [n]$$

$$v_i \tilde{x}_i(v) - t_i(v) \geq v_i(\theta - \alpha) \qquad \forall v \in \mathcal{V}, i \in [n]$$

$$x_i(v) \geq \theta \qquad \forall v \in \mathcal{V}, i \in [n]$$

The first constraint corresponds to *truthfulness* for the IC setting, the second is the IR constraint, and the third is the *obedience* constraint. We have thus reduced the data market design problem to that of Myerson's revenue maximizing single-item auction problem. Instead of $\tilde{x}_i(v)$ denoting the probability of allocating an item, it denotes expected payoff after accounting for the negative externalities. Also, rather than allocative constraints, we have obedience constraints on $x_i(v)$ which requires $x_i(v) \geq \theta$. Thus, by Myerson's theory, the total expected revenue is equal to the expected virtual welfare minus some constant $K$ (stemming from IR constraints) for some $\tilde{x}_i(v_i, v_{-i})$ that is non-decreasing in $v_i$. Thus, we have:

$$\mathbb{E}_{v \sim \mathcal{V}} \left[ \sum_{i \in [n]} [t_i(v_i)] \right] = \mathbb{E}_{v \sim \mathcal{V}} \left[ \sum_{i \in [n]} \phi_i(v_i) \tilde{x}_i(v) \right] - K \tag{36}$$

$$= \mathbb{E}_{v \sim \mathcal{V}} \left[ \sum_{i \in [n]} \phi_i(v_i) \left( x_i(v) - \frac{\alpha}{n-1} \sum_{j \in [n] \setminus i} x_j(v) \right) \right] - K \tag{37}$$

$$= \mathbb{E}_{v \sim \mathcal{V}} \left[ \sum_{i \in [n]} \left( \phi_i(v_i) - \frac{\alpha}{n-1} \sum_{j \in [n] \setminus i} \phi_j(v_j) \right) x_i(v) \right] - K \tag{38}$$

In order to maximize the revenue, we need to maximize the virtual welfare. Thus we can set $x_i(v) = 1$ (a fully informative experiment) when $\phi_i(v_i) - \sum_{j \in [n] \setminus i} \phi_j(v_j) \geq 0$ and we $x_i(v) = \theta$ (an uninformative experiment that always sends the same signal regardless of the state) otherwise. This is precisely the mechanism described in the theorem.

Note that such an $\tilde{x}_i(v_i, v_{-i})$ is non-decreasing in $v_i$ (since $x_i(v_i, v_{-i})$ is non-decreasing in $v_i$ and $x_j(v_i, v_{-i})$ for $j \in [n] \setminus i$ is non-increasing in $v_i$ for regular distributions $\mathcal{V}_i$ and $\mathcal{V}_j$). Moreover, $x_i(v) \geq \theta$ is also satisfied. $\qquad \square$

