# OpenReview forum: "Data Market Design through Deep Learning"
_NeurIPS.cc/2023/Conference — NeurIPS 2023 poster_

### Official Review · Reviewer_nspR · 2023-06-21

**Soundness:** 2 fair
**Presentation:** 2 fair
**Contribution:** 2 fair
**Rating:** 5
**Confidence:** 3

**Summary:**

This paper studies the market design problem, specifically for data markets. In particular, different from existing analytic approaches, the proposed approach is based on (deep) learning to recover/discover market designs. They adopt and extend an existing RochetNet architecture to both single- and multi-buyer setting and empirically demonstrate the effectiveness of the approach in recovering/discovering the market design.

**Strengths:**

- The paper studies the problem of market design and it is relevant for data market.
- The proposed learning-based approach is interesting in that it can recover some analytic solutions.
- There are relatively extensive empirical results.

**Weaknesses:**

- The motivation and justification of a (deep) learning-based approach can be made stronger.

    In lines 40-42, "The difficulty of using analytical tools for this problem of data market design is highlighted by this example, and it remains an open problem to obtain theoretical results for richer multi-buyer settings. This motivates the need for computational approaches." While it is perceived that analytic solutions are difficult, and computational approaches seem a viable alternative. Is it really necessary to use deep learning? In other words, are there less complex computational approaches that can be tried first or reasons why they would not work as well?

    In particular, (how) can the assumption of i.i.d. samples from $\mathcal{P}$ for training the deep learning model be satisfied? It requires the type of the buyer (i.e., both belief and the $v$) to remain fixed throughout observing the signals. Does this assumption have conflicts with "Upon receiving a signal, the buyers update their prior beliefs and choose an optimal action accordingly" (lines 143-144)?


- The inline equations in the paper can break the flow of the writing and make it more difficult for the reader to catch the most important points.

    For instance, equations (1)-(4) are used to discuss (different variants of) incentive compatbility. It is not so clear which equation the reader should pay most attention to. Furthermore, it seems that equation (4) (i.e., ex post incentivie compatible) is not interpreted after the equation.

- Some experimental results can be difficult to interpret (or understand their significance), due to the lack of (existing) analytic characterization of optimum solution.

    For instance, in lines 294-296, "We are aware of no theoretical characterization of optimal data market designs when both $v$ and $\theta$ vary. In such cases, we can use RochetNet to conjecture the structure of an optimal solution." As a result, it is not clear to the reader how to understand whether the proposed method is effective. It further goes to the first point regarding the motivation/justification of a learning based approach: There lacks a solution or ground truth (i.e., analytic optimum or approixmate optimum) to evaluate the approach. Hence, it seems appealing to first establish such a solution before a computational approach, otherwise, how to effectively evaluate the proposed computational approach?


**Questions:**

- In lines 20-22, "... hold vast quantities of data about individuals. In turn, this has led to data markets, where information about an individual can be purchased in real-time to guide decision-making (e.g., LiveRamp, Segment, Bloomreach)." This seems to hint at that the aforementioned companies are selling data about individuals, is it what it means?

- In lines 60-62, "Further, we give a training method that enables the efficient reuse of computed interim allocations and payments from other samples to swiftly calculate the interim utility of misreporting, dramatically speeding up training." Is this empirically or theoretically demonstrated, specifically about "dramatically speeding up training"? What is it comparing against, in terms of speed of training?

- In line 122, "The state of the world, $\omega$, is unknown and is drawn from a finite state space ... " Is there an assumption on the distribution of this?

- In line 127, "where each $v_i$ is drawn independently from a distribution $\mathcal{V}_i$". What is the interpretation of $v_i$ and what does the distribution $\mathcal{V}_i$ depend on?

- In lines 137-138, it seems that the negative externality is in the form of decreasing payment for one buyer $i$ as the gain for some other buyers. In other words, if another buyer $j$ gains (in ex post payoff), this buyer $i$ "loses" (i.e., has a lower utility), is this correct? How should this be interpreted in an example?

- In line 139, "There is a data seller who observes the world state ... " How to justify or realize this assumption that the actual world state is exactly known by the data seller?

- In line 159 (5-th bulletin point), "$u_i(a,\omega, V_i, \theta_i)$", is it meant to be $V_i$ or $v_i$?

- In line 192, "... an unsupervised learning problem." Is it referring to optimizing the softmax version of Equation (9)? If so, it looks more like an optimization problem (i.e., parametric fitting) instead of a learning problem. Often, unsupervised learning is to learn about the inter or intra structure of the data instead of to fit a functional form. Please help interpret why the loss function in line 222 is an unsupervised learning problem.



**Limitations:**

- Typically in an optimization approach, if the objective is non-convex (or more complex), it is difficult to establish theoretical guarantees in terms of the optimality or quality of the final solution obtained. This is also mentioned by the authors in lines 374 - 375. The implication is that, it is difficult to obtained a principled understanding of how good the solution (i.e., learnt market design) is, obtained from the gradient-based optimization.

- With regard to lines 378-380, "we return to where we started, and underline that markets for trading data about individuals raise a number of ethical concerns." In light of the potential ethical concerns of data trading, a (deep) learning-based approach potentially makes it even more difficult to manage and parse the working mechanism of the data trading. As a result, such an approach can make it even more difficult to reliably/verifably address those concerns.

---

> ### Author Rebuttal · Authors · 2023-08-10
>
> **Justification for using Deep Learning:** Please note that the goal here is to learn a differentiable _function_ that represents an entire mechanism, i.e., a complete mapping from all possible inputs to all possible outputs. The goal is not to simply find a pointwise output for a given input because incentive considerations of mechanism designs rely on reasoning about the global properties of a learned function.  Tools from deep learning are very well suited to these kinds of tasks since they are highly configurable and can learn complex and non-linear patterns. In our setting, once we have a differentiable representation of a mechanism and loss function, we can use standard machine learning pipelines to optimize these representations easily (“computation as learning”). As noted, the other added advantage of using a deep learning based approach is the flexibility in accommodating different design requirements.
>
> In regards to the other computational approaches, the most relevant, existing computational method for this data market design problem is Cai and Velegkas (2021), but this focuses on discrete inputs and makes use of linear programming, and does not consider continuous inputs. Handling continuous inputs through discretization and then optimizing for discretized inputs through linear programs (LPs) has been demonstrated to lead to significant scalability issues in the context of computational approaches to the design of optimal auctions Dütting et al. (2019).
>
> In this work, we follow the agenda of differentiable economics and leverage the expressive power of deep neural networks to discover designs that are approximately optimizing and approximately incentive aligned. These approaches have worked well for the settings of auction design, and also two-sided matching problems and problems of social choice, and in this work, we demonstrate that they also perform very well for the design of data markets – with suitable extensions to handle the new behavioral considerations.
>
> **i.i.d assumption:** In this model, all the buyers have a prior belief over which state is likely to occur, and the prior and the payoff for taking the correct action constitutes a buyer’s type. Each buyer then reports their type to the data seller.
>
> From the revelation principle for dynamic games (Myerson 1991), it's sufficient to restrict our attention to mechanisms that are IC (where players report their true types). It’s these types that constitute the training samples from our mechanisms, and thus the assumption of iid samples is satisfied.
>
> A buyer then updates its beliefs if they choose to buy the experiment: they’re sent a signal based on the experiment that they purchase, and this signal is used to update their prior belief. This happens after they’ve taken part in the mechanism, and after the statistical experiments have been sold. Please refer to Ln 152-160 for an overview of the timing of the mechanism.
>
> **Inline equations:** Thanks for pointing this out. We will make the necessary changes to make it more readable.
>
> **Motivation for a learning based approach:** We think about this a little differently – we’re inspired by the challenges with developing theoretical solutions in these multidimensional settings and are looking to the use of computational tools to attack this problem. Can we use this kind of framework to generate interesting conjectures that can then be proved?
>
> This has been a successful pipeline across a number of discovery-based, scientific domains, with in silico design leading to tests in the lab (here, through “the lab” would be “proved” or “tested in the experimental economics lab,” and perhaps “in simulation with AI participants” in the future). One of the main motivations of our work is to provide economists with a new computational tool with which to test conjectures and analyze the structural properties of optimal designs.
>
> ---
> **Companies and Data:**  Indeed, these companies are all customer data platforms which unify first-party customer data from multiple sources to build a single coherent consumer profile. Liveramp, for example, onboards data by matching offline files, names, addresses to digital identifiers so that companies can target users with online ads. Based on this, it aggregates information for a person’s activities from different browsers, devices, and channels that represent a  consumer’s digital footprint and provides access to these data via its marketplace.
>
> **Regarding Speedup:** We will add a note about this. In existing BIC networks, for a batch size B with K samples for computing interim values and M misreports, we will have to compute B x K x M forward passes. However, in our approach, we don’t sample new misreports but rather re-use other data points from the minibatch as misreports, thereby doing only B x K forward passes.
>
> **Assumptions and Explanations:** Some of the assumptions made here are standard in the mechanism design and information design literature. For better clarity, please refer to the practical example discussed in the response to Reviewer u4po [here](https://openreview.net/forum?id=sgCrNMOuXp&noteId=KNtPhdCieC)
>
> **Typos:**  Thanks for pointing this out. It is $v_i$
>
> **Unsupervised learning problem:**  We think of this as unsupervised learning because there are no “ground truth labels,” which in our setting would be examples of how to allocate and price information coming from an optimal mechanism. In fact, the optimal mechanism is a priori unknown in many of the settings studied, and this is a problem of discovery of new mechanisms. Instead, the pipeline that we formulate is used to optimize for revenue subject to IC constraints, and we look to interpret the optimized designs, and introduce new conjectures.
>
> ---
> **Non-Convexity:** Please refer the global comment [here](https://openreview.net/forum?id=sgCrNMOuXp&noteId=KNtPhdCieC).
> **Ethical Concerns:** Please refer to the subcomment.

---

> > ### Author Response · Authors · 2023-08-10
> > **Regarding ethical concerns**
> >
> > Indeed, there are important ethical concerns concerning markets for trading data about individuals, and we can use the additional page in the camera-ready copy to give an expanded discussion.
> >
> > Perhaps most interesting, we expect that the techniques introduced in this paper can also easily be extended to identify market designs that make additional, explicit tradeoffs between welfare and concerns regarding user welfare, including privacy. This can be done by modulating the loss function to incorporate additional considerations, for example, providing a continuous tradeoff between revenue and user (privacy-based or otherwise) welfare. This seems very interesting for future work.
> >
> > In light of the inherent importance of privacy considerations, properly designed and functioning data markets could be an improvement over the many “hidden markets” in the present day, where the quid pro quo trade, for example, between preference information and free content or services, may not be clear to users.
> >
> > We also note that data markets for selling information already exist and are typically subject to regulatory control and that markets for data will likely continue to exist as an important part of the business landscape. Thus, they are worthy of study, and thorough computational approaches that can afford additional considerations from reality relative to theoretical approaches are necessary.

---

> > ### Comment · Reviewer_nspR · 2023-08-17
> > **Reponse to rebuttal**
> >
> > I thank the authors for their detailed feedback and response. Most of my concerns are addressed, so I will raise my rating to 5.
> >
> > Some additional notes:
> > - I do think the deep learning approach is interesting, despite my concerns were over its jusitification as a design choice. I recommend the authors make clear in their revision what they have discussed in the rebuttals (specifically why deep learning, and what it is used for).
> > - Regarding the term "unsupervised learning", I am still a bit unsure whether it's the most apt choice, primarily because unsupervised learning has a conventional meaning attached to it for the machine learning community. Since you do adopt an optimization objective, the name could be something related to optimization and specific to your setting. This is a suggestion.

---

### Official Review · Reviewer_u4po · 2023-07-04

**Soundness:** 3 good
**Presentation:** 2 fair
**Contribution:** 2 fair
**Rating:** 5
**Confidence:** 3

**Summary:**

This paper introduces a deep learning application to the data market designs that find optimal signaling schemes to maximize the revenue of data sellers. The proposed method is designed to handle truthfulness and obedience (i.e., buyers following recommendations). The overall approach follows the prior frameworks of RochetNet and RegretNet for auction design. The authors are able to demonstrate the method’s ability to recover existing analytical optimal solutions and extend to cases where analytical results are not available. Some experimental results are provided for both single-buyer and multiple-buyer settings.

**Strengths:**

1. The paper applies deep learning to the new domain of data market design, illustrating the feasibility of learning solutions to optimal data market design.
2. It considers the obedience of data buyers in the design. This makes the approach more practical.
3. The paper provides a sound analysis of Individual Rationality for the mechanism and payments.

**Weaknesses:**

1. The writing could be improved. Preliminaries could be better structured to explain essential terms like menu entry, signaling, state of the world, how the mechanism works, etc. Interpretations could be added after Lemmas and computation equations (e.g., (10)) to improve clarity.
2. The scales of the experiments are not large enough to be convincing. If larger experiments are not possible, challenges and limitations should be clearly stated.

**Questions:**

**Major**

1. Are there any references to support the assumptions made in the preliminaries section? For example, why is the matching utility payoff reasonable in data market design? How do you interpret that in the binary-state setting in the real world? How about a more complex non-binary setting?
2. For the single buyer setting Lemma 3.1, it is claimed that the mechanism is Incentive Compatible as it is agent optimizing. Why is it agent optimizing when the objective is to maximize the payment by the agents?
3. How to access the validity of the results from the networks when there is no analytical solution (more complex settings)? For example, for the price of 0.14 outputted for setting C, how do you know whether it is close to optimal? Also, could you provide a more intuitive interpretation of the price and results?
4. What are the challenges in conducting experiments on binary states, actions? Also, can you perform experiments on more than two buyers? Can the method be extended to much more complex scenarios with a large number of players, actions and states?

**Minor**

5. Grammar. Lines 80, 103, 242. Punctuations and formats: Lines 146, 153-160, 239.
6. Some notations can be confusing, especially the subscripts, superscripts and brackets.
7. What is $\Delta$ in Line 129, never explained before.

**Limitations:**

The authors have sufficiently discussed the limitations of the approach in the limitation section. Additionally, I wonder how well this framework applies in real-world scenarios. Could the author clarify the limitations of adopting the method in real life for data pricing, or provide a practical example/application?

---

> ### Author Rebuttal · Authors · 2023-08-09
>
> Thanks for the feedback regarding the clarity. We will make the necessary edits and add interpretations to our lemmas and equations to make them more readable. The response to the questions including concerns regarding the scale are addressed below
>
> ---
>
> **1. Matching Utilities under binary states**
> Our focus in this paper is on the model formulated by Bergemann et al. (2018) and Bonatti et al. (2022), where the assumption of matching utility under binary states is widely used. Under this assumption, the buyer faces *binary* payoffs in each state where the outcomes can be nicely categorized into ‘right’ and ‘wrong’. For this case, the restriction is without loss of generality relative to a general payoff matrix. There, for every state (say state 1), we can always subtract the (state-dependent) constant of the utility of taking the non-matching action (action 2) under the current state (state 1). This linear transformation normalizes the payoffs of the data buyer by setting off-diagonal utility to 0 without affecting the optimality conditions of the data buyer's decision problem.  For more details, see the matching utility section in Bergemann et al. (2018). We also discuss a practical example in the last subsection of this comment.
>
> Furthermore, our choice to primarily focus on matching utility is because this is assumed in the results of Bergemann et al. (2018) and Bonatti et al. (2022), which we use as baselines for a comparison of learned and analytically optimum results. The proposed RochetNet and RegretNet formulation only needs a properly specified utility function to work and is readily able to handle custom utility values. The framework can also be easily extended, without new challenges, to a non-binary state problem.
>
> **2. Clarification regarding agent-optimizing**
> We apologize for the lack of clarity. This concept of “agent optimizing” is from the economic theory of mechanism design, and we will add a reference. In particular, it requires that each agent is presented with a menu of options that does not depend on their report, and where their report is – in effect – used to pick an option that maximizes their utility. This is a theoretical framing that allows one to reason about incentive properties. Given this as a design constraint, we can then seek to maximize expected revenue, subject to this property of “agent optimality.”  In the present paper, this “agent picking the best option” is hard-coded in the RochetNet architecture in a differentiable way.
>
> **3. Validity and Interpretation of results**
> For the *ex post IC* setting for the multi-bidder case with uncertain payoffs, we conjecture the structure of the solution and prove its optimality.
>
> For other settings, we conjecture but do not prove the structure of the optimal solution. For Setting C, the conjectured optimal design is a menu of size one that offers a fully informative experiment. For Setting D, we conjecture that the optimal design is a menu of size two with one fully informative experiment and one partially informative experiment. Both these results provide a new target for economic theory.
>
> **4. Scaling up**
> We've addressed this in the global comment [here](https://openreview.net/forum?id=sgCrNMOuXp&noteId=OyHh7ioeDB).
>
> **5, 6, 7. Typos and Clarifications**
> Thanks for pointing this out, and we will address it. The triangle denotes the probability simplex (over states in Ln 129)
>
> ---
>
> **Practical Example and Challenges**
> We adapt the following real-world example from Bonatti et al. (2022)
> Consider a platform similar to Amazon that wishes to monetize selling information about a consumer, like their shopping history, with retailers. The success of each retailer is influenced by two main things: (i) how accurately they can target their ads to consumers, meaning they show the right products to the right people; and (ii) how unique their ads are compared to what other competitors are offering, based on what consumers like.
>
> Retailers decide how much they're willing to pay for extra information based on how much profit they make from each sale, considering their costs. As this is only privately known to the merchant, the platform must elicit it through its choice of mechanism. This reduces to designing a menu of (experiment, payment) pairs, each corresponding to an advertising campaign.
>
> In this setting, there are two states $\Theta$ = {0, 1} representing consumer preference. There are 2 retailers (indexed by $i$ and $j$) whose goal is to match their product to a consumer’s preference. Their action sets are thus given by $a_i = a_j$ = {0, 1}. The payoffs for retailer $i$ is given by $1_{\theta = a_i}  - \alpha \cdot 1_{\theta = a_j}$  where $\alpha$ controls the competition factor. The competitor $j$ having correct information about a consumer increases the competition and imposes a negative externality of $\alpha$ in this case.
>
> *Challenges* - As discussed in Bonatti et al. (2022), this model abstracts away certain details and dynamics of online advertising. For example, the model does not capture the cost of transmitting data, and nor does the model explain a practical style by which partially correct information can be communicated.
>
> ---
>
> *References*
>
> [1] Bergemann, D., Bonatti, A., & Smolin, A. (2018). The design and price of information. American economic review, 108(1), 1-48.
>
> [2] Bonatti, A., Dahleh, M., Horel, T., & Nouripour, A. (2022). Selling information in competitive environments. arXiv preprint arXiv:2202.08780.

---

> > ### Comment · Reviewer_u4po · 2023-08-13
> > **Thanks for the rebuttal**
> >
> > 1. While it is claimed that the choice of “binary states/actions” and “matching utility” is not a necessary assumption for the framework (only adopted due to easy comparison with existing analytical works), no attempts to show the effectiveness of the proposed method on non-binary state problems are made. It is hard to be convinced without concrete support such as experiments.
> >
> > By the way, you mentioned about 10 agents and 10 states experiments in the global response, could you report the results for this experiment? Also, is there no way to assess the quality of the results (since you say visualizing + conjecturing is the way but we cannot do so in this complex case)?
> >
> > 2. Thanks for the clarification on “agent optimizing”, yes, please add a reference.
> >
> > 3. I am not entirely sure about the validity of the conjectures you have made. Further, could you explain about the “**price**” in your experiment outputs? If only one menu option is given, shouldn’t a higher price yield better revenue, which is the overall objective? What should be the optimum?
> >
> > Thanks for the real-world example, too!

---

> > > ### Author Response · Authors · 2023-08-17
> > > **Response**
> > >
> > > Thank you for engaging us on this. We are sharing additional results in support of both scaling-up and models involving non-matching utility. We will also use the additional page in the final version of the paper to add this analysis. We break the results into those for the _ex post_ IC setting and the Bayesian IC (BIC) setting.
> > >
> > > ### Scaling
> > > For the results on scaling, check the comment [here](https://openreview.net/forum?id=sgCrNMOuXp&noteId=yemOZcWgh5)
> > >
> > > ### Non-matching utilities
> > > Extensions to the results in the current paper to handle the case of non-matching utilities are simple because this only involves changing the values in the payoff matrix.
> > >
> > > As an illustration, we report here on a replication of the results for a setting with non-matching utilities that is reported in Bergemann et al. [1]. They make a case for complex designs, showing that a seller offering a single, fully-informative experiment can achieve only an O(m) approximation to optimal revenue in settings with $m \geq 3$ actions. We illustrate this in the case of a single buyer, with binary world state, 4 actions, and prior belief $(\theta, 1 - \theta)$ with $\theta$ sampled from a uniform [0, 1] distribution, and payoffs given by this matrix $\mathcal{U} = [[1,0.9, 0.6, 0],[0,0.5,0.7,1]]$.
> > >
> > > |      **Menu**      | **Revenue** |
> > > |------------------|:-----------:|
> > > | Fully Informative  |    0.111    |
> > > |      RochetNet     |    **0.119**   |
> > >
> > > By training a mechanism with our pipeline, we confirm that selling a single, fully-informative experiment yields lower revenue than the menu learned by RochetNet, which involves selling a fully-informative experiment along with a partially-informative experiment with $\pi =$
> > > [[ 0, 0, 0, 1], [ 0, 0,  0.6,  0.4]].
> > >
> > > ### Why we can't simply increase the price
> > > In this work, we seek to maximize revenue while incentivizing truthful behavior from buyers. Increasing the price associated with an experiment involves a tradeoff. On one hand, improving revenue when selling the experiment, while on the other hand this may push buyers away from the experiment in favor of purchasing another experiment or declining to purchase information altogether.
> > >
> > > To illustrate this tradeoff, consider the following simple scenario: the matching utility case and a single buyer whose payoff is uniform on [0, 1]. In this case, the optimal (_ex post_) IC mechanism offers a fully informative experiment at a price of 0.5. The buyer opts to purchase if the expected value of information from the experiment is larger than 0.5.  Given the buyer’s prior, this happens with probability 0.5, resulting in a revenue to the seller of 0.25. However, at a higher price of 0.6, the probability of purchase falls to 0.4, giving lower revenue of $0.6 \times 0.4 = 0.24$.
> > >
> > > ### Validity of Conjectures
> > > We refer to Appendix C.2 for additional experiments where we study how the differential informativeness of experiments varies with properties such as the distribution of belief types of buyers, including the precision of buyer beliefs. Understanding this kind of structure between economic primitives and optimal market design is interesting within economic theory, as demonstrated by [2], who examined this setting for discrete types and established analytical solutions. In our experiments, we develop support with this new framework for analogous conjectures in the setting of continuous types. For additional illustrations, our framework also allows us to identify scenarios where a more sophisticated menu can outperform a market that sells a single, fully-informative experiment — in Section 5; we showcase instances where our framework learns to sell an additional, partially-informative experiment to boost revenue. Again, this provides a target for economic theory.
> > >
> > > ---
> > > *References*
> > > [1] Bergemann, D., Cai, Y., Velegkas, G., & Zhao, M. (2022, July). Is Selling Complete Information (Approximately) Optimal?. In Proceedings of the 23rd ACM Conference on Economics and Computation (pp. 608-663).
> > >
> > > [2] Bergemann, D., Bonatti, A., & Smolin, A. (2018). The design and price of information. American economic review, 108(1), 1-48.

---

> > > > ### Comment · Reviewer_u4po · 2023-08-21
> > > >
> > > > I would like to thank the authors for the additional illustration on the extensions and the conjectures. I am happy to keep my scores, leaning toward the acceptance of this paper.

---

### Official Review · Reviewer_wNRV · 2023-07-05

**Soundness:** 4 excellent
**Presentation:** 2 fair
**Contribution:** 3 good
**Rating:** 6
**Confidence:** 4

**Summary:**

The authors are concerned with a problem of "data market design". In such a setting, a mechanism designer with access to an unknown world state interacts with buyers who have private types, and need to take actions whose payoffs vary depending on the world state. These buyers purchase (in the single buyer case) or bid on (in the multi-buyer case) access to a signaling scheme which, given reports from the agents and the world state, sends a signal to the buyers (which without loss of generality can just be a recommended action). This mechanism, treated as a direct-revelation mechanism, needs to be both truthful (incentivizing honest reporting by the buyers) and obedient (once the buyers receive their signal, they should be incentivized not to deviate from the recommendation). Subject to those constraints (either Bayesian or ex post), the mechanism designer wants to maximize their revenue.

This problem shares some similarities to truthful revenue-maximizing auction design. In that domain, there has been recent progress using the tools of "differentiable economics" to approximately learn high-performing (and sometimes even provably optimal) auctions, in both single- and multi-bidder settings.

The authors apply very similar techniques to this data market problem. In single-buyer settings (as in auctions) they are able to ensure exact IC; for multi-buyer settings they use a Lagrangian during training to approximately enforce IC constraints. They experiment on a relatively wide variety of problem instances, reproducing known results, finding new optimal mechanisms, and conjecturing optimal mechanisms where they cannot find them.

**Strengths:**

The paper comprehensively shows how to successfully apply differentiable economics to a new domain where it has not previously been applied. The authors are able to reproduce optimal mechanisms and find new ones, showing that their adaptation of these technique is in fact useful in producing novel results. This helps to further push these techniques towards being practically helpful tools for theorists and modelers.

**Weaknesses:**

The network architectures here are essentially the same as those used in previous work for auctions, only adapted slightly for the data market setting. This is fine, but it does mean that from the perspective of differentiable economics, there is no novel methodological contribution.

The experiments appear to consider at most 2 buyers. While (as in the case of multi-parameter auctions) even selling to just two buyers may be a very challenging case, it would be more interesting to consider a slightly larger number of buyers.

**Questions:**

Can the method in fact scale to larger (even just 3-5 buyers) settings, or not? This should be discussed.

**Limitations:**

See questions.

---

> ### Author Rebuttal · Authors · 2023-08-09
>
> Thanks for the review and the feedback. We've discussed concerns regarding the weaknesses - especially regarding **Methodological Contributions** and **Scaling up** in the global comment [here](https://openreview.net/forum?id=sgCrNMOuXp&noteId=OyHh7ioeDB).

---

> > ### Comment · Reviewer_wNRV · 2023-08-17
> > **Acknowledging rebuttal**
> >
> > Thanks for your response here. I acknowledge that the BIC techniques do constitute a new methodological contribution, although this is not the main point of the paper. Thanks also for checking that scaling up can generally work, at least for DSIC. My general conclusion about this paper has not changed.

---

### Official Review · Reviewer_QSAL · 2023-07-06

**Soundness:** 3 good
**Presentation:** 3 good
**Contribution:** 3 good
**Rating:** 5
**Confidence:** 3

**Summary:**

This paper introduces a deep learning framework for the automated design of data markets, a novel and timely application in the field of economics. The authors address the data market design problem, which involves designing a set of signaling schemes to maximize expected revenue. The paper extends previous work on deep learning for auction design by learning signaling schemes and handling obedience constraints that arise from the actions of agents.

**Strengths:**

- Innovative Application: The paper introduces a novel application of deep learning to the data market design problem, expanding the scope of machine learning in the field of economics.

- The paper is well-written overall.

**Weaknesses:**

-Incremental work: It seems that the core contribution, the proposed neural network architecture, is a simple extension of existing model called RochetNet, by slightly modifying the loss function.

-Lack of comparison with baselines: mechanism design for information acquisition is a long standing problem. I was surprised to see no baseline comparison in the experiments, and no discussion on how/why existing approaches may not work in the methodology.

**Questions:**

What are some baseline methods to compare with? For example, how does the standard rochetnet perform on the proposed market settings?

**Limitations:**

Yes.

---

> ### Author Rebuttal · Authors · 2023-08-09
>
> **Novelty**
>
> We've addressed this in the global comment [here](https://openreview.net/forum?id=sgCrNMOuXp&noteId=OyHh7ioeDB).
>
> ---
>
> **Comparison with other baselines**
>
> We are unaware of other computational baselines, and we already compare them with the theoretically optimal designs from [1, 2] where they are available.  In terms of prior computational results, we primarily deal with continuous distributions, whereas [3] provides computational results for discrete types. Handling continuous distributions through discretization using linear programs (LPs) is understood to lead to significant scalability issues, as demonstrated for optimal auction design [4].
>
> We also discuss some alternate data market formulations in the related work section. But these are different problems, and solution approaches to these problems do not provide baselines for the problem formulation that we take up.
>
> RochetNet cannot be used for the data market setting, since this departs from optimal auction design in several substantive ways. The incentive compatibility constraints under the data market setting are more complex, comprising both the truth-telling part (reporting one’s type truthfully) and the obedience part (taking the recommended action instead of deviating to another action). If we adopt the standard RochetNet idea from the auction setting, we will miss the obedience part, assuming agents act according to our action recommendation when they might deviate to other actions strategically instead.
>
> Problems of information design, particularly “Bayesian Persuasion,” have been studied in both economics and computer science, but this is a quite different setting from that of the present paper. In particular, it does not model problems of revenue, bidding, or multiple receivers (or competition across multiple receivers).
>
> Problems of optimal auction design are also widely studied in computer science and economics, but very different in structure to the data market problem; e.g., much of the auction literature considers the sale of one or more “rival good” (i.e., a good that cannot be allocated to multiple people, as is the case with information). Moreover, the auction design literature does not deal with posterior beliefs, arising from information sale, and effect of beliefs on downstream actions.
>
>
> ---
>
> *References*
>
> [1] Bergemann, D., Bonatti, A., & Smolin, A. (2018). The design and price of information. American economic review, 108(1), 1-48.
>
> [2] Bonatti, A., Dahleh, M., Horel, T., & Nouripour, A. (2022). Selling information in competitive environments. arXiv preprint arXiv:2202.08780.
>
> [3] Cai, Y., & Velegkas, G. (2021). How to sell information optimally: An algorithmic study. 12th Innovations in Theoretical Computer Science Conference (ITCS 2021), pages 81:1--81:20.
>
> [4] Dütting, P., Feng, Z., Narasimhan, H., Parkes, D., & Ravindranath, S. S. (2019, May). Optimal auctions through deep learning. In International Conference on Machine Learning (pp. 1706-1715). PMLR.

---

### Official Review · Reviewer_zvsA · 2023-07-07

**Soundness:** 4 excellent
**Presentation:** 3 good
**Contribution:** 3 good
**Rating:** 6
**Confidence:** 2

**Summary:**

The authors present a novel approach to the problem of data market design, which seeks to find a set of signaling schemes, each revealing some of the information known to a seller and having a corresponding price, where the goal is to maximize expected revenue. Then, the authors introduce the application of a deep learning framework to the automated design of the data market. The paper discusses the importance of data market design and its potential applications in real-world settings, such as data marketplaces where sellers sell data to buyers for ML tasks. The authors demonstrate that their new learning framework can replicate known solutions from theory, expand to more complex settings, and establish the optimality of new designs. The paper also highlights some limitations of the approach, such as the need for interpretability of the mechanisms learned by the RegretNet approach for larger problems, the potential for local optima in non-convex problems, and the challenge of achieving exact incentive alignment in multi-buyer settings.

**Strengths:**

+ The paper presents a novel approach to the problem of data market design, which uses deep learning to automate the design of data markets.

+ The authors demonstrate that their new learning framework can almost precisely replicate all known solutions from theory, which shows that the approach is effective and reliable.

+ The paper shows that the new learning framework can be used to establish the optimality of new designs and conjecture the structure of optimal designs, which is a significant contribution to the field.



**Weaknesses:**

+ The paper acknowledges that for the approach to provide insights into the theoretically optimal design for larger problems, it will be important to provide interpretability to the mechanisms learned by the approach. However, the RegretNet approach used in the paper is not immediately interpretable, which limits its usefulness in this regard.

+ The paper notes that the approach uses gradient-based approaches, which may suffer from local optima in non-convex problems. This suggests that the approach may not always find the global optimum and may be limited in its ability to handle more complex problems.

+ The paper attains in the multi-buyer setting approximate and not exact incentive alignment, which leaves the question as to how much alignment is enough for agents to follow the intended advice of a market design. This suggests that the approach may not be able to achieve exact incentive alignment in all settings, which could limit its effectiveness.


**Questions:**

+ Could you provide more details on how the RegretNet approach can be made more interpretable for larger problems? Are there any specific techniques or methods that could be used to achieve this?

+ Have you considered using other optimization techniques besides gradient-based approaches to address the potential for local optima in non-convex problems? If so, what are some alternative approaches that could be used?

+ What are some potential ways to provide more practical or theoretical guidance on how much alignment is enough for agents to follow the intended advice of a market design? Are there any existing frameworks or approaches that could be used to address this issue?


**Limitations:**

The authors acknowledge the ethical concerns raised by markets for trading data about individuals and suggest that machine learning frameworks such as those introduced in this paper can be used to strike new kinds of trade-offs, such as allowing individuals to benefit directly from trades on data about themselves. This shows that the authors are aware of the broader implications of their work and are thinking critically about its potential impact.

---

> ### Author Rebuttal · Authors · 2023-08-09
>
> **Interpretability**
>
> We can interpret the designs learned by RegretNet in a few ways.
> - We can plot the probability of recommending the correct action for different agent types. We adopt this approach and visualize these as heatmaps in the main paper (Figure 2) and Appendix (Figures 8, 9, and 10) to show different optimal designs. This kind of visualization borrows from the existing economics literature and can be extended for larger designs.
> - We can compare the similarity of outputs learned by RegretNet to well-known baselines and analyze how economic properties of interest vary with different design choices. For instance,  [1, 2] study how externalities affect the optimal design of data markets. In our paper, as an illustrative example, we show how we can understand revenue patterns for new settings, studying here the effect of varying the intensity of negative externality on revenue.
> - Another interesting direction, not taken here but interesting for future work, is to initialize the mechanism to a known baseline, use our framework to optimize it further and see whether known mechanisms can be improved.
> - Another interesting direction for future work is to consider techniques such as distillation to “compile” a design into something more interpretable. Generally, we like that this agenda is “future proof,” meaning that advances in machine learning, including distillation and interpretability, can be applied here.
>
> ---
> **Non-Convex Formulations and Alternate Optimization Methods**
>
> We've addressed this in the global comment [here](https://openreview.net/forum?id=sgCrNMOuXp&noteId=OyHh7ioeDB).
>
> ---
> **Approximate Incentive Alignment**
>
> First, we emphasize that we can achieve very small empirical regret violations (see Fig. 11 and Fig. 15 in the Appendix). We find this encouraging. Another useful observation, in support of the effectiveness of approximate incentive alignment, comes from the machine learning pipeline almost exactly recovering optimal designs for all known settings (as per the above comments).
>
> That said, it remains an empirical question as to how much alignment “is enough.” We would expect this to depend on context, e.g., how complex it is to deviate from truthful reporting vs. satisficing behavior, how high the stakes are, and how well-informed the participants are. An interesting path forward in this regard would be to develop simulators of behavior as an additional way to test performance. There is also a growing literature on modeling complexity in the context of strategic paper, e.g., [3].
>
> In the context of auction design, there is also some recent guiding theory [4, 5, 6] that provides transformations between $\epsilon$-BIC and BIC without revenue loss. Interesting avenues for future work include extending these transformations to general class problems — problems with both types and actions, as in the data market settings. It also remains open to extending these results to obtain $\epsilon$-DSIC to DSIC transformations for general distributions.
>
> ---
>
> *References*
>
> [1] Agarwal, A., Dahleh, M., Horel, T., and Rui, M. (2020). Towards data auctions with externalities. arXiv preprint arXiv:2003.08345
>
> [2] Bonatti, A., Dahleh, M., Horel, T., & Nouripour, A. (2022). Selling information in competitive environments. arXiv preprint arXiv:2202.08780.
>
> [3] Modibo K. Camara. 2022. Computationally Tractable Choice. In Proceedings of the 23rd ACM Conference on Economics and Computation (EC '22)
>
> [4] Daskalakis, C., & Weinberg, S. M. (2012, June). Symmetries and optimal multi-dimensional mechanism design. In Proceedings of the 13th ACM conference on Electronic commerce (pp. 370-387).
>
> [5] Cai, Y., Oikonomou, A., Velegkas, G., & Zhao, M. (2021). An Efficient∊-BIC to BIC Transformation and Its Application to Black-Box Reduction in Revenue Maximization. In Proceedings of the 2021 ACM-SIAM Symposium on Discrete Algorithms (SODA) (pp. 1337-1356). Society for Industrial and Applied Mathematics.
>
> [6] Conitzer, V., Feng, Z., Parkes, D.C., Sodomka, E. (2022). Welfare-Preserving $\epsilon$-BIC to BIC Transformation with Negligible Revenue Loss. In: Feldman, M., Fu, H., Talgam-Cohen, I. (eds) Web and Internet Economics. WINE 2021.

---

> > ### Comment · Reviewer_zvsA · 2023-08-13
> >
> > I appreciate the authors' responses.
> >
> > I am wondering in the case of malicious bidders, when most of the bidders are colluding. Would the seller be then underpaid?

---

> > > ### Author Response · Authors · 2023-08-16
> > >
> > > Collusion is indeed a valid concern in auction design and one that can lead to underpayment. Although our setting is more complex, which may make it harder to sustain collusive agreements between bidders, it could still be a problem in the present setting. Moreover, Collusion and bid rigging are legally prohibited in many contexts under regulations such as the Sherman Antitrust Act [2]. Considering this, we follow the majority of the auction- and mechanism design literature and focus on individual strategy proofness.
> > >
> > > The theoretical economics literature does also formalize the concept of collusion resistance, as well as the weaker concept of _group strategyproofness_, which considers deviations by groups of bidders and differ as to whether or not they support side payments between bidders.  In auction settings, the requirement of collusion-resistance is in fact very strong and essentially equivalent to requiring take-it-or-leave-it prices [3] (in the other directions, there is some theory to identify when single-agent SP is sufficient for group SP).
> > >
> > > To our knowledge, collusion considerations have never been formally studied in the context of the design of data markets.  Certainly, it would be interesting to extend our notions of regret to allow for _group regret_, suitably defined and studied in the context of our machine-learning pipeline. Indeed, this pipeline allows for intermediate definitions, for example, considering deviations by groups of a limited size.
> > >
> > > ---
> > >
> > > *References*:
> > >
> > > [1] Barberà, S., Berga, D., & Moreno, B. (2010). Individual versus group strategy-proofness: When do they coincide?. Journal of Economic Theory, 145(5), 1648-1674.
> > >
> > > [2] U.S. Department of Justice. (2016). Sherman Antitrust Act. Retrieved from https://www.justice.gov/d9/pages/attachments/2016/01/05/211578.pdf
> > >
> > > [3] A. V. Goldberg, J. D. Hartline, Collusion-resistant mechanisms for single-parameter agents, Proc. SODA '05: Proceedings of the sixteenth annual ACM-SIAM symposium on Discrete algorithms 2005, pp. 620–629

---

> > > > ### Comment · Reviewer_zvsA · 2023-08-21
> > > >
> > > > Thank you for your insights. I think it would be important to extend the work to measure the resistance to collusion for a more realistic data market setting. I will keep my positive score for this paper.

---

### Author Rebuttal · Authors · 2023-08-09

We thank all the reviewers for their helpful feedback. Three main comments are regarding non-convexity of our formulation, novelty and scaling up. We address these concerns here. The remaining questions and concerns are addressed in the individual responses.

---
**Non-Convex Formulations and Alternate Optimization Methods**

Although local optima can be a concern when optimizing non-convex objective functions, gradient-based methods have demonstrated remarkable success, with extensive empirical evidence showing convergence to high-quality solutions and even in the presence of non-convexity. There is also theoretical support for a “no local optima” phenomenon in the ML theory literature. In addition, a recent paper [1] also shows how the local optima are connected by a path, where the revenue along the path is at least as much as one of the endpoints, justifying the empirical success of RochetNet for auction design. Although open, we conjecture similar theoretical results for the adapted version of RochetNet to the present paper.

Regarding our specific experiments, one way in which we validate the effectiveness of the proposed method is by looking at settings in which theoretical results are available. In this way, we demonstrate that we can reliably recover optimal solutions for all prior settings in which analytical solutions are known (see the continuum of types setting in Section 5 and the BIC settings in Section 6)

Considering alternate computational approaches for the problem that we study, while [2] propose using LPs for settings with discrete types, we don’t expect this to extend to the continuous settings of the present paper since introducing discretization to enable the application of LPs leads to significant scalability challenges, as demonstrated in [3] in application to revenue-optimal auction design.

---

**Novelty**

We respectfully disagree that attaining the ability to train models in this pipeline is “just about modifying the loss function” from earlier work. Rather, the crucial aspect is to capture a new kind of strategic behavior within the learning pipeline.  A significant innovation is to be able to handle both _obedience_ and _truthfulness_ constraints on behavior, including _double deviations_ (misreporting, along with deviating from suggested actions). To our knowledge, this is novel in the differentiable economics literature.

We also introduce new sampling techniques for achieving BIC, which are able to reuse interim allocations and interim payment calculations to compute IC violations. In existing BIC networks [4], for batch size, B with K samples for computing interim values and M misreports, we will have to compute B x K x M forward passes. By contrast, in our approach, we don’t sample new misreports but rather reuse other data points from the minibatch as misreports, thereby doing only B x K forward passes. This substantially speeds up the training process.

Another contribution is the new results we provide in the application domain itself, which is an application domain garnering considerable interest in economics and computer science and one with substantial societal importance, given the multifaceted nature of the design problem.  Whereas analytical results are only available for the BIC setting, which is, in effect, lower-dimensional, and easier to analyze, we are able to study through computational techniques the design of data markets in the _ex post_ IC setting, which is a setting without existing theory. In Section 6, for the _ex post_ IC setting, we conjecture the structure of the optimal designs and prove their optimality through Myerson's framework (proof in the Appendix). We see this as an important contribution, as _ex post_ IC is a stronger notion of IC than BIC.

In addition, we provide an illustrative example to showcase the framework's versatility as a toolbox for economists. For instance, in Section 6, we study how the revenue varies as we vary the intensity of negative externality, thereby varying the competition. This hasn’t been studied before for the setting with uncertain priors, and we show economically meaningful variation, as no analytical solution is known.

---

**Scaling up**

We choose to study the case of two buyers because it is easier to visualize the results and develop conjectures. We would also like to emphasize that Theorem 6.1, which is stated for the *ex post* IC setting with uncertain payoffs, holds for any number of bidders.

The described approach can scale to more buyers and states, especially in the ex post IC settings. For instance, in the case of uncertain priors, for 10 agents and 10 states, our approach takes 62 min to run for 20000 iterations on a single NVIDIA Tesla V100 GPU.

Scaling is harder for the BIC setting, however. As we increase the number of agents to $n$, obtaining the interim values accurately involves computing the marginal values over $n  - 1$ dimensions. This can be quite expensive.

We will add a discussion on scaling to the paper.

---

*References*

[1] Hertrich, C., Tao, Y., & Végh, L. A. (2023). Mode Connectivity in Auction Design. arXiv preprint arXiv:2305.11005.

[2] Cai, Y., & Velegkas, G. (2021). How to sell information optimally: An algorithmic study. 12th Innovations in Theoretical Computer Science Conference (ITCS 2021), pages 81:1--81:20.

[3] Dütting, P., Feng, Z., Narasimhan, H., Parkes, D., & Ravindranath, S. S. (2019, May). Optimal auctions through deep learning. In International Conference on Machine Learning (pp. 1706-1715). PMLR.

[4] Feng, Z., Narasimhan, H., & Parkes, D. C. (2018, July). Deep learning for revenue-optimal auctions with budgets. In Proceedings of the 17th International Conference on Autonomous Agents and Multiagent Systems (pp. 354-362).

---

> ### Author Response · Authors · 2023-08-17
> **On Scaling**
>
> As noted in the above comment during this rebuttal, our approach scales well in the _ex post_ IC setting but does not scale as easily with the number of buyers in the BIC setting. The challenge in the BIC setting comes from the "interim computations" $–$ involving conditional expectations over reports of others.  For a given buyer and report, inference in regard to the "interim experiment" involves computing an expectation over the reports of the remaining buyers. For this reason, we only report results for 2 and 5 buyers in the BIC case (in regard to scaling the number of world states in the BIC case, there is a technical challenge preventing this in our current code framework but not an in-principle challenge).  Scaling beyond this will require new techniques, for example, exploiting symmetry.
> Please note that the *ex post* IC case is in some ways more interesting because it provides an equilibrium concept that does not rely on common knowledge on the part of buyers about buyer types. This makes the learned mechanisms more robust (e.g., changes to the type distribution or deviations in participants' beliefs).  Also noteworthy is that existing theory only provided results in the BIC case, which turns out to be easier to handle theoretically because the analytical problem becomes, in effect, smaller in dimension.
>
> Note that the revenue achieved in some cases, in reporting these new empirical results, is slightly larger than the optimal revenue because of the small empirical ex post regret.
>
> ### Ex post IC
>
> **Binary States with Matching Utility, $\theta = (0.5, 0.5)$, and $\alpha = 0.5$**
> | Num Buyers | Optimal revenue | Learned revenue | Empirical ex post Regret | Time Taken |
> |:----------:|:---------------:|:---------------:|:------------------------:|:----------:|
> |      3     |       0.39      |       0.40      |          <0.001          | 11 minutes |
> |      5     |       0.64      |       0.65      |          <0.001          | 21 minutes |
> |     10     |       1.26      |       1.27      |          <0.001          | 34 minutes |
>
>
>
> The optimal revenue is computed through the optimal mechanisms from Theorem 6. The learned revenue and regret are from the multi-buyer architecture discussed in the paper with the same set of hyper-parameters in Appendix B.4
>
> In the submitted paper, we conjecture and prove the structure of the optimal mechanism in the _ex post_ IC setting for binary world states, matching utility, and for any number of buyers. We can extend the proof for the matching utility case to non-binary states (and will extend the paper accordingly). We use this new theory to compute the optimal revenue in Column 1.
>
> **10 States with Matching Utility, $\theta = (0.1, ..., 0.1)$, and $\alpha = 0.5$**
> | Num Buyers | Optimal revenue | Learned revenue | Empirical ex post Regret | Time Taken |
> |:----------:|:---------------:|:---------------:|:------------------------:|:----------:|
> |      3     |       0.70     |       0.72     |          <0.001          | 16 minutes |
> |      5     |       1.15      |       1.16      |          <0.001          | 28 minutes |
> |     10     |       2.27      |       2.22      |          <0.001          | 64 minutes |
>
> ### Bayes IC
> **Binary States with Matching Utility, $\theta = (0.5, 0.5)$, and $\alpha = 0.5$**
> | Num Buyers | Optimal revenue | Learned revenue | Empirical ex post Regret | Time Taken |
> |:----------:|:---------------:|:---------------:|:------------------------:|:----------:|
> |      3     |       0.52      |       0.52     |          <0.001          | 13 minutes |
> |      5     |       0.78      |       0.82      |          <0.001          | 40 minutes |

---

### Decision · Program_Chairs · 2023-09-21

**Decision:**

Accept (poster)

**Comment:**

The submitted paper considers a deep learning framework for automated data market design that can handle obedience and truthfulness constraints and experimentally shows that the framework can replicate known solutions from theory but also extend to novel designs. The paper was reviewed by 5 knowledgeable reviewers, all of whom recommended acceptance (3x borderline, 2x weak). They appreciated the novel application and the proposed deep-learning framework. A discussion among reviewers and authors took place, and the authors were able to clarify many of the raised concerns and provide additional details regarding raised questions. The discussions revealed that the submitted paper has shortcomings in clearly explaining certain aspects of the proposed framework (scalability, novelty relative to some existing works, parts of the experimental evaluation) but the authors provided clear and convincing responses therefor. Hence, despite the shortcomings, I am recommending acceptance of the paper but strongly encourage the authors to improve their paper in line with the reviewers' comments, the discussions, and the clarifications provided by the authors themselves.